# CURE: A Pre-training Framework on Large-scale Patient Data for Treatment Effect Estimation

## Abstract

Treatment effect estimation (TEE) refers to the estimation of causal effects, and it aims to compare the difference among treatment strategies on important outcomes. Current machine learning based methods are mainly trained on labeled data with specific treatments or outcomes of interest, which can be sub-optimal if the labeled data are limited. In this paper, we propose a novel transformer-based pre-training and fine-tuning framework called **CURE** for TEE from observational data. **CURE** is pre-trained on large-scale unlabeled patient data to learn representative contextual patient representations, and then fine-tuned on labeled patient data for TEE. We design a new sequence encoding for longitudinal (or structured) patient data and we incorporate structure and time into patient embeddings. Evaluated on 4 downstream TEE tasks, **CURE** outperforms the state-of-the-art methods in terms of an average of 3.8% and 6.9% absolute improvement in Area under the ROC Curve (AUC) and Area under the Precision-Recall Curve (AUPR), and 15.7% absolute improvement in Influence function-based Precision of Estimating Heterogeneous Effects (IF-PEHE). We further demonstrate the data scalability of **CURE** and verify the results with corresponding randomized clinical trials. Our proposed method provides a new machine learning paradigm for TEE based on observational data.

## 1 Introduction

Treatment effect estimation (TEE) is to evaluate the causal effects of treatment strategies on some important outcomes, which is a crucial problem in many areas such as healthcare (Glass et al., 2013), education (Dehejia & Wahba, 1999) and economics (Imbens, 2004). Randomized clinical trials (RCTs) are the *de-facto* gold standard for identifying causal effects through randomizing the treatment assignment and comparing the responses in different treatment groups. However, conducting RCTs is time-consuming, expensive and sometimes unethical. Observational data such as medical claims provide a promising opportunity for treatment effect estimation when RCTs are expensive or impossible to conduct.

Recently, many works have been proposed to adopt neural networks (NNs) for TEE from observational data (Shalit et al., 2017; Shi et al., 2019; Hassanpour & Greiner, 2019; Curth & van der Schaar, 2021b;a; Zhang et al., 2022b; Guo et al., 2021). Compared to classical TEE methods such as regression trees (Chipman et al., 2010) or random forests (Wager & Athey, 2018), NN-based methods achieve better performance in handling the complex and nonlinear relationships among covariates, treatment and outcome. However, there are still some common limitations of existing TEE methods: 1) Most model designs are task-specific or data-specific so it is hard to adapt the model to a more generalized setting. 2) Existing labeled dataset often has small-scale data size, whereas training neural models requires large and high-quality labeled data for capturing inherent complex relationships of the input data.

Recently, Transformer (Vaswani et al., 2017) has been widely adopted as a critical and unified building block in the pre-training and fine-tuning paradigm across data modalities. The pre-trained Transformer-based models (PTMs) have become the model of choice in many deep learning domains such as natural language processing (NLP) (Devlin et al., 2018; Radford et al., 2018; 2019;

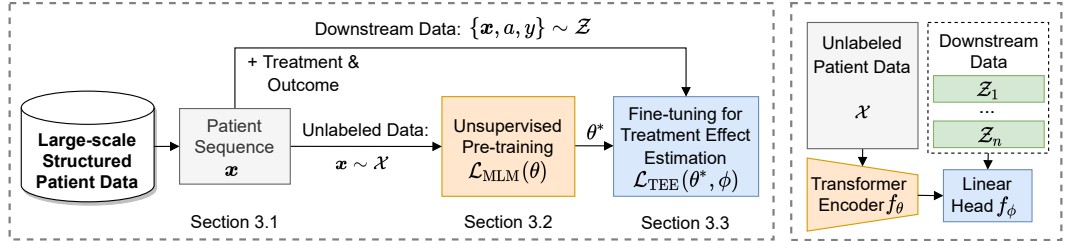

Figure 1: The overall pipeline of **CURE**. It mainly consists of three parts: 1) data encoding of longitudinal patient data; 2) unsupervised pre-training on unlabeled data and 3) fine-tuning on downstream labeled data for treatment effect estimation. In TEE, labels mean the studied treatment $\alpha$ and outcome $y$ of the patient sequence $\boldsymbol{x}$.

Brown et al., 2020; Liu et al., 2019) and computer vision (CV) (Carion et al., 2020; Dosovitskiy et al., 2020; Parmar et al., 2018). The dominant approach is to pre-train on a large-scale dataset with unsupervised or self-supervised learning and then fine-tune on a smaller task-specific dataset. Nonetheless, applying this pre-training and fine-tuning paradigm to treatment effect estimation problems faces the following three major challenges: 1) encoding structured longitudinal observational patient data into sequence input; 2) lack of well-curated large-scale pre-training dataset; 3) lack of real-world downstream treatment effect estimation tasks to benchmark baselines.

In this paper, we propose a new pre-training and fine-tuning framework for estimating causal effect of a treatment: **Ca**Usal t**R**eatment **E**ffect estimation (**CURE**). As shown in Fig. 1, the large-scale structured patient data are extracted from a real-world medical claims data (MarketScan Research Databases [1]). We first encode the structured data as sequential input by chronologically flattening and aligning all observed covariates. We obtain around 3M processed unlabeled patient sequences for pre-training. And the downstream datasets with labeled treatment and outcome are created according to specific TEE tasks from established RCTs. Based on the retrospective study design and domain knowledge, we obtain 4 downstream tasks and each of them containing 10K-20K patient samples. The task is to evaluate the comparative effectiveness of two treatment effects in reducing the risk of stroke for patients with coronary artery disease (CAD). Second, we pre-train a Transformer-based model on the unlabeled data with an unsupervised learning objective to generate contextualized patient representations. To accommodate the issues of complex hierarchical structure (i.e., the patient record contains multiple visits and each visit contains multiple types of medications or diagnoses) and irregularity of the observational patient data, we propose a comprehensive embedding method to incorporate the structure and time information. Finally, we fine-tune the pre-trained model on various downstream TEE tasks.

We are the first study to demonstrate the success of adopting the pre-training and fine-tuning framework to representation learning of patient data for TEE, together with necessary but minimal changes on the transformer architecture design, and real-world case studies on randomized clinical trials. We summarize our **main contributions** as follows.

- We propose **CURE**, a novel transformer-based pre-training and fine-tuning framework for TEE. We present a new patient data encoding method to encode structured observational patient data and incorporate covariate type and time into patient embeddings.

- We obtain and preprocess large-scale patient data from real-world medical claims data as our pre-training resource. We derive 4 downstream TEE tasks according to study designs and domain knowledge from established RCTs for model evaluation.

- We conduct thorough experiments and show that **CURE** yields superior performance on all downstream tasks compared to state-of-the-art TEE methods. We achieve, on average, 3.8% and 6.9% absolute improvement in AUC and AUPR respective for outcome prediction, and 15.7% absolute improvement in IF-PEHE for TEE over the best baseline among 4 tasks. We also verify the estimated treatment effects with the conclusion of corresponding RCTs.

- We further explore the effectiveness of **CURE** in several ablation studies including the proposed patient embedding, the influence of pre-training data size on downstream tasks, and the generalizability of low-resource fine-tuning data.

---

[1]https://www.ibm.com/products/marketscan-research-databases

## 2 BACKGROUND AND RELATED WORK

**Treatment effect estimation from observational data.** In this paper, we are interested in observational patient data. Each patient sample consists of pre-treatment covariates $\boldsymbol{x}$ (i.e., historical co-medication, co-morbidities and demographics) and a treatment $a$ of interest. Following the potential outcome framework (Rubin, 2005), the potential outcome $y_a$ is defined as the response to treatment $a$ out of all available treatment options. Typically, we consider the comparative treatment effects of two treatments and denote two potential outcomes as $y_1$ and $y_0$ for simplicity.

We aim to estimate the individual-level treatment effect (ITE) as the difference between the potential outcomes under two treatment arms as $y_1(\boldsymbol{x}) - y_0(\boldsymbol{x})$. We are also interested in the average treatment effect (ATE) which is the average effect among the entire population, denoted as $\mathbb{E}[y_1(\boldsymbol{x}) - y_0(\boldsymbol{x})]$. In observational data, only one of the potential outcomes is available and the remaining counterfactual outcomes are missing in nature, which makes this task more difficult than classical supervised learning. We follow the standard assumptions (Imbens & Rubin, 2015) (i.e., consistency, positivity and strong ignorability). The potential outcome can be defined as $y_a(\boldsymbol{x}) = \mathbb{E}[y|a, \boldsymbol{x}]$, and can be estimated from observational data. More details of assumptions are illustrated in Appendix A.

**Deep learning for treatment effect estimation.** Generally, existing NN-based methods formulate the TEE as several regression tasks (i.e., regression on potential outcomes and treatment) with different levels of information shared among the nuisance estimation tasks using representation learning. TARNet (Curth & van der Schaar, 2021b), for example, learns one shared representations for two potential outcomes, while SNet (Curth & van der Schaar, 2021b) learns five different representations on the combinations of treatment and potential outcomes. Recently, Transformer has been introduced as an encoder block for TEE (Zhang et al., 2022b; Guo et al., 2021) and yields better performance compared to the state-of-the-art methods. Despite the promising results, the main limitation is that the model performance can be diminished if the labeled dataset is limited. The model trained for one particular problem or data may fail to generalize to other scenarios.

**Pre-train and fine-tune of Transformer.** Since Transformer is based on a flexible architecture with few assumptions on the input data structure, it is difficult to directly train the model on small-scale data. Therefore, various pre-trained Transformer-based models (PTMs) are first pre-trained on the large-scale unlabeled data and then fine-tuned for labeled tasks at hand. PTMs learn universal and contextualized representations, which can boost various downstream tasks, and avoid developing and training a new model from scratch. Among the existing PTMs in NLP, BERT (Devlin et al., 2018) is one of the most popular models. BERT (Devlin et al., 2018) is pre-trained on large-scale unlabeled corpus via self-supervised pre-training tasks (i.e., masked language modeling and next sentence prediction) and fine-tuned on downstream tasks with an additional linear head. Our observational patient data are close to natural language text as they both contain sequential information. However, patient data have some unique characteristics that distinguish them from the text. Compared to the text, patient data contain a more complex hierarchical structure and time information. Therefore, existing BERT pre-training architecture can not be directly applied to modeling patient data.

## 3 CURE: A PRETRAINING AND FINE-TUNING FRAMEWORK FOR TEE

In this section, we introduce our **CURE** framework (as shown in Fig. 1) which includes three key steps: (1) Encoding structured patient data as sequential input by aligning medications and diagnosis in each visit chronologically (Sec. 3.1), (2) Pre-training on a large-scale unlabeled patient data by minimizing the unsupervised objective and obtaining the optimized parameters $\theta^*$ (Sec. 3.2), and (3) Fine-tuning on a small-scale labeled downstream dataset for TEE by jointly optimizing $\theta^*$ and a linear head parameterized as $\phi$ (Sec. 3.3).

### 3.1 ENCODING STRUCTURED PATIENT DATA

In this work, we focus on longitudinal observational patient data. We first introduce the data for pre-training and fine-tuning respectively. Then we illustrate how to convert structured patient data into sequential input for the Transformer encoder.

**Pre-train data structure.** The pre-training is based on large-scale unlabeled patient data. Here, to distinguish from downstream data, we denote the pre-train data as unlabeled data ($\boldsymbol{x} \sim \mathcal{X}$), while

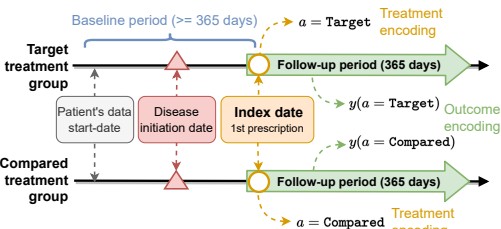

Figure 2: Illustration of the downstream data construction with retrospective study design. Index date refers to the first prescription of the target treatment or the compared treatment, which should be no prior to the disease initiation date. The baseline period is no less than one year and the follow-up period as outcome observation is also one year. The treatments of interests and outcomes are obtained at the index date and follow-up period respectively.

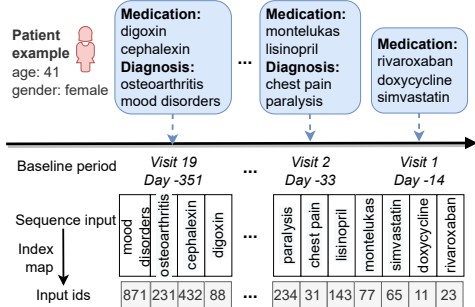

Figure 3: Illustration of encoding structured patient data into sequential input. The patient data are recorded in an hierarchical structure such that a patient contains multiple visits and each visit contains multiple medications and diagnoses. The structured data are converted into a sequence by flattening all covariates in each visit and aligning them chronologically.

downstream data with treatment $a$ and outcome $y$ as labeled data ($\{\boldsymbol{x}, a, y\} \sim \mathcal{Z}$). The unlabeled patient data consist of: (1) Co-medication $m_1, m_2, ..., m_{|\mathcal{M}|} \in \mathcal{M}$, where $|\mathcal{M}|$ is the number of unique medication names. (2) Co-morbidities $d_1, d_2, ..., d_{|\mathcal{D}|} \in \mathcal{D}$, where $|\mathcal{D}|$ is the number of unique diagnosis codes. (3) Demographics $c$: age encoded as categorical value and gender encoded as binary value. A patient can have multiple visits $\{v_1, \ldots, v_T\}$, where each of visit $v_t$ contains a subset of medication and diagnosis codes ($v_t \in \mathcal{M} \cup \mathcal{D}$). We denote the unlabeled patient data as $\boldsymbol{x} = \{c, \{v_t\}_{t=1}^T\}$ and all the covariates are obtained from the baseline period as shown in Figure 2. We build a medical vocabulary from all patient covariates as $\mathcal{V} = \{\mathcal{M}, \mathcal{D}, c\}$.

**Fine-tune data structure.** The fine-tuning is based on a small-scale labeled patient data, which are not used for pre-training. Besides the co-medication, co-morbidities and demographics, the labeled patient data contain treatment $a \in \mathcal{M}_{\text{task}}$ (i.e., can be either the target treatment or compared treatment from task-specific medication group $\mathcal{M}_{\text{task}}$) and outcome $y_a \in \{0, 1\}$ under the observed treatment $a$. In Figure. 2, we show the retrospective study design of how to construct downstream data and obtain labels for treatments and outcomes. Specially, we collect patient data from two different treatment groups for comparison. For each group of patients, the covariates (i.e., co-medication, co-morbidities and demographics) are obtained from the baseline period (a.k.a., pre-treatment covariates) as potential confounders, and the outcomes are obtained from the follow-up period. More illustrations of the study design can be found in Appendix C.

**Structured patient data to sequential input.** As introduced above, the original patient data are recorded naturally in a hierarchical structure. Unlike natural language text, which is inherently encoded as a sequence of words, the patient data need to be preprocessed into a "sequence-like" format before sending to the Transformer encoder. As shown in Fig. 3, we flatten the structured patient data by chronologically going through each medication and diagnosis in each visit and aligning them in one sequence. Each medication or diagnosis is encoded as an individual token, which is comparable to text tokenization. The token ids are obtained from the medical vocabulary $\mathcal{V}$.

### 3.2 PRE-TRAINING **CURE**

As shown in Fig. 4, the pre-training consists of three modules: (1) an embedding layer to convert input patient data into embedding representations, (2) Transformer encoders to generate contextualized hidden representations and (3) a final project layer for pre-training objective. More formally, given the encoded patient sequence $\boldsymbol{x} = [x_1, \ldots, x_m, \ldots, x_T]$ as specified in Sec. 3.1, the pre-training procedure can be decomposed into the following steps:

$$\boldsymbol{x} \xrightarrow{\text{Mask}} \left[x_1, \ldots, [\texttt{MASK}]_m, \ldots, x_T\right] \xrightarrow{\text{Embedding}} \{\boldsymbol{e}_i\}_{i=1}^T \xrightarrow{f_\theta} \{\boldsymbol{h}_i\}_{i=1}^T \xrightarrow{\text{MLM}} \mathcal{L}_{\text{MLM}}(\theta) \quad (1)$$

We randomly replace 15% of input tokens with special $[\texttt{MASK}]$ tokens, e.g., token $x_m$ in the sequence. $\boldsymbol{e}_i \in \mathbb{R}^B$ denotes the embedding representation with embedding dimension $B$ generated by the comprehensive embedding layer. $\boldsymbol{h}_i \in \mathbb{R}^H$ denotes the contextualized representation with hidden dimension $H$ generated by Transformer encoder $f_\theta$. The masked language modeling (MLM)

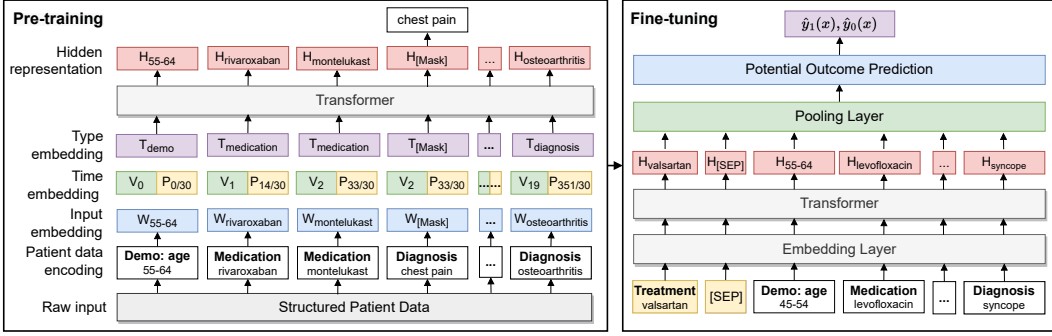

Figure 4: Illustration of pre-training and fine-tuning of **CURE**. The unlabeled structured patient data are first converted into a sequential input, and processed for the embedding layer and encoder. During the fine-tuning, the treatments of interest are appended for potential outcome prediction. (Devlin et al., 2018) aims to predict the masked tokens $x_m$ from the established vocabulary $\mathcal{V}$ using hidden representation $\boldsymbol{h}_m$. The pre-training loss function of MLM is denoted as $\mathcal{L}_{\mathrm{MLM}}(\theta)$ with optimization parameters $\theta$.

**Comprehensive embedding layer.** The pre-trained language models like BERT (Devlin et al., 2018) have achieved great success in natural language text demonstrating strong power in modeling sequential data. Though longitudinal patient data can be deemed as a kind of sequential data when organized by the chronological order, there exist substantial and unignorable discrepancies between the text and patient data. It is hard to directly apply the existing pre-trained language model to our unique patient data. Our ablation study shows that the standard embedding design adopted in NLP (i.e., token embedding and position embedding) will be sub-optimal in our scenario (see Sec. 4.3 for more details).

Compared to the natural language text, (1) longitudinal patient data contain a more complex hierarchical structure than the text data: a patient record contains a number of visits and each visit also contains a number of different types of medical codes (i.e., medication or diagnosis). (2) The patient data are irregularly sampled (i.e., the time interval among visits is not regular) while the text data are regularly organized. As shown in Fig. 3, the visit dates are not regularly distributed along the time: the first visit happened on day 0, the second visit happened on day 14 and the third one on day 33, etc. On visit 2 (day 33), the patient received two types of codes: montelukas and lisinopril as medications, and chest pain and paralysis as diagnoses.

To accommodate the above issues of complex hierarchical structure and irregularity of the observational patient data, we propose a more comprehensive embedding layer than the original BERT (Devlin et al., 2018) embedding layer by including associated code type information and time information. For each input token, the patient embedding $\boldsymbol{e}_i$ is obtained as:

$$\boldsymbol{e}_i = \boldsymbol{w}_{\mathrm{token}} + \boldsymbol{t}_{\mathrm{type}} + \boldsymbol{v}_{\mathrm{visit}} + \boldsymbol{p}_{\mathrm{physical}} \quad (2)$$

where $\boldsymbol{w}_{\mathrm{token}}$ is the original input token embedding. $\boldsymbol{t}_{\mathrm{type}}$ is the type embedding of the input token. According to our data, there are three types in total: {Demographics, Medication, Diagnosis}. The visit time embedding $\boldsymbol{v}_{\mathrm{visit}}$ is the visit time corresponding to a visit. The physical time embedding $\boldsymbol{p}_{\mathrm{physical}}$ is the physical time associated with the visit. Here, the physical time is measured by month (i.e., 30-day fixed window). Both visit and physical time are organized relative to the treatment index date (i.e., the absolute distance between the visit/physical time to the index date).

As an illustration, in Fig. 4, the input is a sequence of patient data containing the type and time information: rivaroxaban is from Medication type prescribed on visit 1 (day 14) and chest pain is from the Diagnosis type received on visit 2 (day 33). The input token embedding, time embedding and type embedding are integrated and used as the input to the Transformer encoder.

**Transformer encoder and pre-training objective** We use an N-stacked Transformer as our encoding backbone as it has been a widely adopted architecture. For each single Transformer encoder block, it consists of a multi-head self-attention layer followed by a fully-connected feed-forward layer (Vaswani et al., 2017). More details of the Transformer architecture are in Appendix B.

The Transformer encoder $f_\theta$ takes the comprehensive embedding representations as input and generates contextualized hidden representations as $f_\theta(\boldsymbol{e})$. Given unlabeled patient data $\mathcal{X}$, the pre-training is to minimize the MLM loss of predicting the masked token with position $j \in \mathcal{J}$ using the input

token embedding and hidden representation:

$$\mathcal{L}_{\text{MLM}}(\theta) = \mathbb{E}_{\boldsymbol{x} \sim \mathcal{X}} \Big[ - \sum_{j \in \mathcal{J}} \log(\text{P}(\boldsymbol{w}_j | f_\theta(\boldsymbol{e}))) \Big] \tag{3}$$

where $\text{P}(\boldsymbol{w}_j | f_\theta(\boldsymbol{e}))$ is the softmax probability of the masked token over all tokens in the vocabulary.

### 3.3 FINE-TUNING **CURE** FOR TEE

Given downstream labeled data $\{\boldsymbol{x}, a, y\} \sim \mathcal{Z}$, we fine-tune the model on different downstream TEE tasks. Here, we are interested in the comparative causal treatment effect of the target treatment over another compared treatment according to the downstream tasks. For each task, we plug in the task-specific input and outputs into **CURE**. We add a linear head $f_\phi$ to the hidden representations learned from the pre-training stage. We fully fine-tune all model parameters end-to-end by jointly updating $\theta^*$ obtained from optimizing Eq. 3 and a randomized $\phi$.

Specially, we append the original input sequence with the index treatment (i.e., target treatment or compared treatment) which is separated by the special [SEP] token. As shown in Fig. 4, the treatment "valsartan" is appended to the original inputs to indicate that the patient is from the treatment group of "valsartan". The model processes the new inputs through the embedding layer and the Transformer encoder with parameters initialized with $\theta^*$. We use the final hidden vector corresponding to the first input token ([CLS]) as the pooled representation $\boldsymbol{h}_{\text{[CLS]}}$ from the pooling layer. We predict the potential outcomes under the treatment $a$ via the linear head as $f_\phi \circ f_{\theta^*}(\boldsymbol{h}_{\text{[CLS]}}(a))$. The fine-tuning objective is the binary cross entropy (BCE) of the potential outcome prediction:

$$\mathcal{L}_{\text{TEE}}(\theta^*, \phi) = \mathbb{E}_{\{\boldsymbol{x}, a, y\} \sim \mathcal{Z}}[\text{BCE}(f_\phi \circ f_{\theta^*}(\boldsymbol{h}_{\text{[CLS]}}(a)), y)] \tag{4}$$

Here, only the factual outcome are used for training loss computation as the counterfactual outcomes are unavailable in the observational data. After model fine-tuning, we infer the ITE $\delta$ and ATE $\Delta$ as the difference between two predicted potential outcomes under the target and compared treatment:

$$\hat{\delta} = \hat{y}_{a=\text{Target}} - \hat{y}_{a=\text{Compared}}; \quad \hat{\Delta} = \mathbb{E}[\hat{y}_{a=\text{Target}} - \hat{y}_{a=\text{Compared}}] \tag{5}$$

## 4 EXPERIMENTS

In this section, we evaluate the proposed **CURE** from three aspects: 1) Quantitative analysis of the comparison performance with state-of-the-art TEE methods on 4 downstream tasks; 2) Qualitative analysis including the validation of the estimated treatment effects with corresponding RCTs, and self-attention feature weights visualization; 3) Ablation studies including proposed feature embedding, pre-training data size, and generalizability of low-resource fine-tuning data.

**Pre-training data.** We extract patient data from MarketScan Commercial Claims and Encounters (CCAE) [2] from 2012 to 2017, which contains individual-level, de-identified healthcare claims information from employers, health plans and hospitals. In this paper, we evolve patients who have ever been diagnosed with coronary artery disease (CAD) as our disease cohort. The definition of CAD is in Appendix C. After conducting data preprocessing and study design, we obtain 2,955,399 patient sequences for pre-training. We obtain 9,435 medical codes including 282 diagnosis codes (i.e., we map the original ICD-9/10 billing codes into Clinical Classifications Software [CCS] [3]) and 9,153 medication codes (i.e., we map medications based on generic names from RED BOOK [4]).

**Downstream tasks.** As the ground truth treatment effects are not available in observational data, we use RCTs as the gold standard to verify our results. We focus on CAD-related RCTs which study the comparative effectiveness of two treatments for reducing the risk of stroke after CAD. We first collect all available Phase 2 and Phase 3 RCTs with CAD as disease name and stroke as disease outcome from https://clinicaltrials.gov/. Stroke is selected because it is commonly used as the primary outcome measurement in various CAD studies and it is well-defined in observational data. Then we select completed RCTs with published results. Finally, we end up with 4 RCTs that meet all the above criteria. We derive corresponding downstream tasks from our data based on the study design as specified in Fig. 2. More details of screening RCTs are in Appendix C. An additional semi-synthetic setting with comparison results are in Appendix D.

---

[2]https://www.ibm.com/products/marketscan-research-databases
[3]www.hcup-us.ahrq.gov/toolssoftware/ccs/ccs.jsp
[4]https://www.ibm.com/products/micromedex-red-book

**Baselines.** We compare **CURE** with 8 neural network models for TEE, including state-of-the-art methods. For models designed for continuous outcomes with mean square error (MSE) as a training objective, we change the objective function to binary cross entropy for consistency. All the baselines are only trained on downstream data and are summarized below:

• *TARNet* (Shalit et al., 2017) predicts the potential outcomes based on balanced representations among treated and controlled groups.

• *DragonNet* (Shi et al., 2019) jointly optimizes treatment prediction and potential outcome prediction. The model first learns shared representations given the input data and then does prediction tasks via a three-head neural network: one for treatment prediction and two for potential outcomes.

• *DR-CFR* (Hassanpour & Greiner, 2019) learns disentangled representations for counterfactual regression and assumes that the observed covariates can be disentangled into three components: only contributing to treatment selection, only contributing to outcome predication, and both.

• *TNet* (Curth & van der Schaar, 2021b) is a neural network based T-learner (Künzel et al., 2019) (i.e., a type of meta-learners that decomposes the TEE into two sub-regression problems). TNet adopts two neural models as base learners for two potential outcomes.

• *SNet* (Curth & van der Schaar, 2021b) also learns disentangled representations and assumes that the observed covariates can be disentangled into five components by considering two potential outcomes separately.

• *FlexTENet* (Curth & van der Schaar, 2021a) incorporates the idea of inductive bias for the shared structure of two potential outcomes into TEE. The model adaptively learns what to share between the potential outcome functions.

• *TransTEE* (Zhang et al., 2022b) is a recently proposed Transformer-based TEE model. The covariates and treatments are encoded via a Transformer and aggregated for outcome prediction via a cross-attention layer.

• *Base Model* directly trains on the downstream datasets using the same architecture as **CURE**.

**Metrics.** We evaluate the factual prediction performance using the standard classification metrics: Area under the ROC Curve (AUC) and Area under the Precision-Recall Curve (AUPR). We evaluate the counterfactual prediction performance using the influence function-based precision of estimating heterogeneous effects (IF-PEHE) (Alaa & Van Der Schaar, 2019), which helps to benchmark TEE methods when the ground truth effects are not available. Compared to the widely adopted precision of estimating heterogeneous effects (PEHE) that measures the mean squared error between estimated treatment effects and true treatment effects, IF-PEHE measures the mean squared error between estimated treatment effects and approximated true treatment effects. The output of the IF-PEHE metric is a numeric value and the lower the better. More details of this metric are in Appendix C.

**Implementation details.** Our pre-training uses the BERT$_{base}$ architecture (Devlin et al., 2018) with 768 hidden size, 12 attention heads, 12 layer Transformer and 3072 intermediate size. The maximum input sequence length is 256. The pre-training is conducted on 3 NVIDIA GeForce RTX 2080 Ti 11GB GPUs with a batch size of 96. We train our model using the adaptive moment estimation (Adam) optimizer, with an initial learning rate of $1e-4$ and learning rate warmup in the first 10% training steps. During the fine-tuning, the learning rate is $5e-5$ without learning rate warmup. We fine-tune the model on each task for 2 epochs. The downstream data are randomly split into training, validation and test sets with percentages of 90%, 5%, 5% respectively. All results are reported on the test sets. More implementation details are mentioned in Appendix C. The code of our proposed **CURE** is available in Supplementary material.

## 4.1 QUANTITATIVE ANALYSIS

**Comparison with state-of-the-art methods.** Table 1 shows the performance of factual outcome prediction (measured by AUC and AUPR) and TEE (measured by IF-PEHE) on four different downstream tasks. We compare **CURE** with the state-of-the-art TEE methods and report the results under 20 random runs. We observe that the proposed **CURE** has more than 3.8%, 6.9% and 15.7% respective average AUC, AUPR and IF-PEHE improvement over the best baseline on these tasks. The results illustrate the promise and effectiveness of our proposed pre-training and fine-tuning methodology for TEE. Notably, even without pre-training, the base model of **CURE** attains similar performance as the best baseline, which suggests the effectiveness of our architecture and data encoding designs. We demonstrate the model performance on addressing the treatment selection bias from quantitative and qualitative perspectives in Appendix D.

Table 1: Comparison with state-of-the-art methods on four downstream datasets. The results are the average and standard deviation over 20 runs.

| Method | Rivaroxaban v.s. Aspirin | | | Valsartan v.s. Ramipril | | |
| --- | --- | --- | --- | --- | --- | --- |
| | AUC ↑ | AUPR ↑ | IF-PEHE ↓ | AUC ↑ | AUPR ↑ | IF-PEHE ↓ |
| TARNet | $0.719 \pm 0.015$ | $0.327 \pm 0.023$ | $0.546 \pm 0.044$ | $0.683 \pm 0.028$ | $0.263 \pm 0.029$ | $0.545 \pm 0.061$ |
| DragonNet | $0.757 \pm 0.013$ | $0.381 \pm 0.023$ | $0.715 \pm 0.082$ | $0.683 \pm 0.026$ | $0.263 \pm 0.027$ | $0.533 \pm 0.109$ |
| DR-CFR | $0.759 \pm 0.015$ | $0.381 \pm 0.026$ | $0.653 \pm 0.118$ | $0.751 \pm 0.019$ | $0.333 \pm 0.032$ | $0.642 \pm 0.135$ |
| TNet | $0.715 \pm 0.016$ | $0.318 \pm 0.028$ | $0.500 \pm 0.059$ | $0.673 \pm 0.021$ | $0.256 \pm 0.024$ | $0.494 \pm 0.065$ |
| SNet | $0.756 \pm 0.014$ | $0.380 \pm 0.028$ | $0.247 \pm 0.054$ | $0.752 \pm 0.022$ | $0.333 \pm 0.033$ | $0.526 \pm 0.097$ |
| FlexTENet | $0.717 \pm 0.014$ | $0.319 \pm 0.022$ | $0.565 \pm 0.054$ | $0.662 \pm 0.028$ | $0.241 \pm 0.027$ | $0.602 \pm 0.088$ |
| TransTEE | $0.717 \pm 0.011$ | $0.299 \pm 0.021$ | $0.557 \pm 0.042$ | $0.773 \pm 0.013$ | $0.380 \pm 0.025$ | $0.427 \pm 0.089$ |
| Base Model | $0.758 \pm 0.029$ | $0.406 \pm 0.037$ | $0.180 \pm 0.038$ | $0.780 \pm 0.050$ | $0.365 \pm 0.066$ | $0.190 \pm 0.065$ |
| **CURE** | $\mathbf{0.803 \pm 0.011}$ | $\mathbf{0.469 \pm 0.023}$ | $\mathbf{0.173 \pm 0.038}$ | $\mathbf{0.811 \pm 0.018}$ | $\mathbf{0.428 \pm 0.041}$ | $\mathbf{0.158 \pm 0.062}$ |

| Method | Ticagrelor v.s. Aspirin | | | Apixaban v.s. Warfarin | | |
| --- | --- | --- | --- | --- | --- | --- |
| | AUC ↑ | AUPR ↑ | IF-PEHE ↓ | AUC ↑ | AUPR ↑ | IF-PEHE ↓ |
| TARNet | $0.714 \pm 0.008$ | $0.359 \pm 0.016$ | $0.520 \pm 0.048$ | $0.748 \pm 0.012$ | $0.447 \pm 0.030$ | $0.535 \pm 0.043$ |
| DragonNet | $0.741 \pm 0.009$ | $0.397 \pm 0.020$ | $0.433 \pm 0.096$ | $0.792 \pm 0.018$ | $0.519 \pm 0.035$ | $0.461 \pm 0.095$ |
| DR-CFR | $0.745 \pm 0.007$ | $0.403 \pm 0.021$ | $0.580 \pm 0.108$ | $0.798 \pm 0.015$ | $0.531 \pm 0.032$ | $0.503 \pm 0.073$ |
| TNet | $0.709 \pm 0.009$ | $0.360 \pm 0.020$ | $0.490 \pm 0.061$ | $0.741 \pm 0.015$ | $0.432 \pm 0.032$ | $0.519 \pm 0.039$ |
| SNet | $0.742 \pm 0.008$ | $0.400 \pm 0.020$ | $0.298 \pm 0.053$ | $0.795 \pm 0.014$ | $0.525 \pm 0.034$ | $0.414 \pm 0.054$ |
| FlexTENet | $0.710 \pm 0.010$ | $0.351 \pm 0.015$ | $0.487 \pm 0.046$ | $0.735 \pm 0.012$ | $0.413 \pm 0.033$ | $0.578 \pm 0.030$ |
| TransTEE | $0.747 \pm 0.022$ | $0.385 \pm 0.015$ | $0.387 \pm 0.021$ | $0.799 \pm 0.011$ | $0.517 \pm 0.031$ | $0.409 \pm 0.059$ |
| Base Model | $0.751 \pm 0.025$ | $0.425 \pm 0.04$ | $0.206 \pm 0.031$ | $0.791 \pm 0.029$ | $0.539 \pm 0.039$ | $0.251 \pm 0.045$ |
| **CURE** | $\mathbf{0.793 \pm 0.008}$ | $\mathbf{0.489 \pm 0.024}$ | $\mathbf{0.198 \pm 0.068}$ | $\mathbf{0.826 \pm 0.014}$ | $\mathbf{0.588 \pm 0.024}$ | $\mathbf{0.224 \pm 0.066}$ |

## 4.2 QUALITATIVE ANALYSIS

**Validate with RCT conclusion.** As the ground truth treatment effects are not available in observational data, we further evaluate the estimated treatment effects with corresponding ground truth RCTs. In Table 4.2, we show the confidence intervals of estimated effects under 20 runs and RCT conclusions of each downstream task.

Table 2: Comparison of the estimated treatment effects with corresponding ground truth RCT. The estimated effects are shown in $95\%$ confidence intervals (CI) under 20 bootstrap runs. The RCT conclusions are obtained from published articles.

| Target v.s. Compared | Estimated Effect (CI) | P value | Generated Hypothesis | RCT Conclusion |
| --- | --- | --- | --- | --- |
| Rivaroxaban v.s. Aspirin | [-0.009, 0.006] | 0.452 | No significant difference | No significant difference (Anand et al., 2018) |
| Valsartan v.s. Ramipril | [-0.003, 0.014] | 0.103 | No significant difference | No significant difference (Pfeffer et al., 2021) |
| Ticagrelor v.s. Aspirin | [0.022, 0.040] | 6e-14 | T. is less effective than A. | No significant difference (Sandner et al., 2020) |
| Apixaban v.s. Warfarin | [-0.039, -0.002] | 4e-4 | A. is more effective than W. | A. is more effective than W. (Granger et al., 2011) |

We use the direct difference to estimate the treatment effects (Hernán, 2004). The results can be interpreted as two potential conclusions: 1) The target treatment is significantly more effective than the compared treatment in reducing the risk of the outcome if the upper bound of the confidence interval is lower than zero. (2) The target is not significantly more effective than the compared treatment if the confidence interval covers zero (i.e., no significant difference) or the lower bound is higher than zero (i.e., the compared treatment is more effective than the target treatment). As we can see, our estimated treatment effects are mostly consistent with each corresponding RCT conclusion. Though the generated hypothesis and RCT conclusion are not exactly the same for the third pair (Ticagrelor v.s. Aspirin), they both indicate that there is no significant reduced treatment effect of the target treatment over the compared treatment. The results demonstrate that our proposed **CURE** successfully identifies correct treatment effects using only observational patient data. The full results of all the baselines can be found in Appendix D.

**Self-attention visualization.** The self-attention mechanism of the Transformer enables the exploration of interaction among input covariates and provides a potential interpretation of the prediction results. We use a Transformer visualization tool called bertviz (Vig, 2019) to help visualize learned attention weights. We show the visualization results of some patient samples in Appendix D.

## 4.3 ABLATION STUDIES

**Effect of embedding layer.** We evaluate the effect of proposed time embedding (visit time and physical time) and type embedding respectively. As shown in Fig. 5, the model with both time and type embedding generally performs better than the other two embedding ablations. Especially,

incorporating time embedding yields larger performance improvement than the type embedding. This indicates that the proposed embedding method is better than the standard embedding method and time information plays a more important role in TEE than the type information.

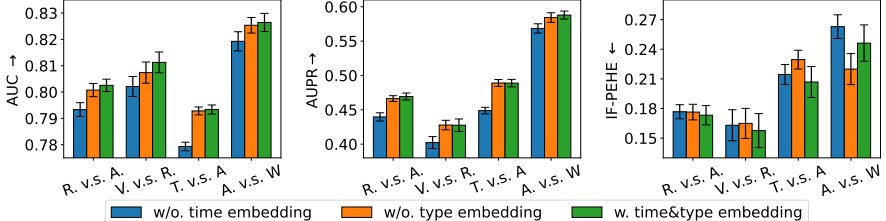

Figure 5: The effect of different embedding layer designs on four downstream tasks.

**Effect of downstream data size.** We demonstrate the model's effectiveness on the low resource of downstream data in Fig. 6. The plots show the model performance with different fractions of labeled downstream data. Generally, given only 5% 10% labeled data, the **CURE** achieves comparable performance to the Base Model which is trained on the fully labeled data. Specifically, the performance gains are large when given a small fraction of labeled data (1%-5%) and the curve tends to gently increase after the fraction is larger than 10%. With increased data size, the performance gradually achieves the upper bound of fine-tuning on fully labeled data. The results demonstrate that unsupervised pre-training benefits low-resource downstream tasks even when only a limited number of labeled data are available for fine-tuning. More results of other metrics are in Appendix D.

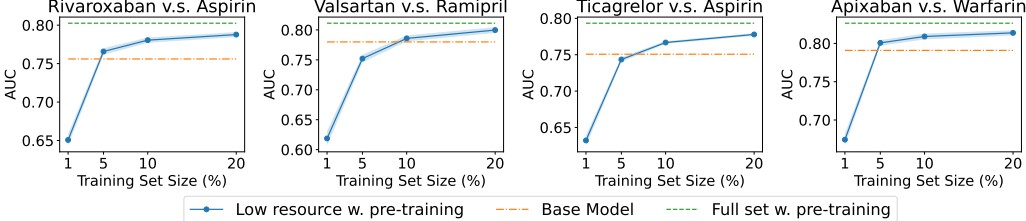

Figure 6: The effect of low resource in fine-tuning datasets on four downstream tasks with different fractions of labeled training set (x-axes). The results are the average of 20 runs.

**Effect of pre-training data size.** We further explore the effect of pre-training data volume on the performance of downstream tasks. In Fig. 7, we show the AUC given different fractions of pre-training data. Here, 0% training set size denotes the Base Model, which is trained on the downstream data from scratch. Generally, the performance improves with the increase of pre-train data. The results indicate that pre-training is beneficial for downstream tasks by learning contextualized patient representations from large-scale unlabeled patient data. Other metrics are show in Appendix D.

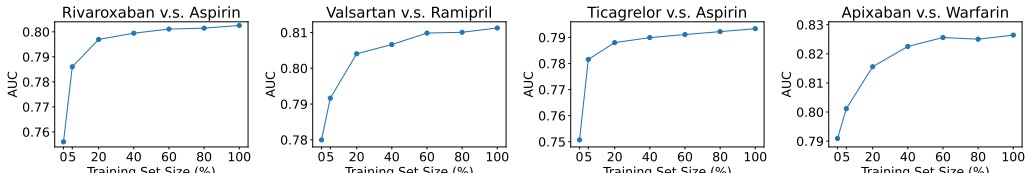

Figure 7: The effect of pre-training data volume on four downstream tasks (average of 20 runs).

## 5 CONCLUSION

In this paper, we study the problem of TEE from observational data. We propose a new transformer-based TEE framework called **CURE**, which adopts the pre-training and fine-tuning paradigm. **CURE** is pre-trained on large-scale unlabeled patient data and then fine-tuned on labeled patient data for TEE. We convert the structured patient data into sequence and design a new sequence encoding method to encode the structure and time into a comprehensive patient embedding. Thorough experiments show that pre-training significantly boosts the TEE performance on 4 downstream tasks compared to state-of-the-art methods. We further demonstrate the data scalability of **CURE** and verify the results with corresponding published RCTs.

**Ethics Statement.** The observational data used in the paper are from MarketScan Research Database, which is fully HIPAA-compliant de-identified, have very minimal risk of the potential for loss of privacy. Moreover, Per the DUAs with MarketScan, all users to access the data will need to take full research, ethics, and compliance training courses and be covered by IRBs. Thus, potential privacy and security risk would be eliminated and/or mitigated.

**Reproducibility Statement.** We provide the code and instructions needed to reproduce the results in supplemental material. The experimental data can be obtained from `https://www.ibm.com/products/marketscan-research-databases`.

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

APPENDIX

## A  CAUSAL ASSUMPTIONS

We follow the standard causal assumptions (Imbens & Rubin, 2015) to help guarantee that the treatment effects are identifiable from the observational data.

**Assumption 1 (Consistency)** *The potential outcome under the treatment $a$ equals to the observed outcome if the actual treatments is $a$.*

**Assumption 2 (Positivity)** *Given the observational data of the history, if the the probability $P(a = 1|\boldsymbol{x}) \neq 0$, then the probability of receiving treatment 0 or 1 is positive, i.e., $0 < P(A = a|X = \boldsymbol{x}) < 1$, for all $a \in \mathcal{A}$ and $\boldsymbol{x} \in \mathcal{X}$.*

**Assumption 3 (Strong Ignorability)** *Given the observational data of the history, the treatment assignment is independent of the potential outcome, i.e., $Y(A = a) \perp\!\!\!\perp A|X = \boldsymbol{x}$, for all $a \in \mathcal{A}$.*

Assumption 1 is fundamental to the potential outcome framework used to define counterfactuals and infer treatment effects. Essentially, this assumption requires that the treatment specified in the study must be precise enough that any variation within the treatment specification will not lead to a different outcome. Assumption 2 implies that all patients may receive the treatment whatever their observed covariates. Otherwise, it is impossible to derive the counterfactuals for patients who do not have any chance of being in the other treatment group. Assumption 3 states that the potential outcomes are independent of treatment assignment given the set of observed covariates. This assumption guarantees that the treatment effects are identifiable given the treatment, outcome and observed covariates as: $\mathbb{E}[Y(A = 1) - Y(A = 0)] = \mathbb{E}_{\boldsymbol{x} \in \mathcal{X}}[\mathbb{E}[Y|A = 1, \boldsymbol{x}] - \mathbb{E}[Y|A = 0, \boldsymbol{x}]]$.

**Positivity in our data.**    We investigate the positivity assumption in each downstream data respectively. Following the positivity evaluation in Shimoni et al. (2019), we estimate the propensity score of each individual via a propensity score model. The positivity can be evaluated through the distribution of propensity: whether the distribution is generally smooth with propensities normally distributed instead of much accumulation in either propensity equals 0 or propensity equals 1. As shown in Figure A1, the estimated propensity scores in each downstream dataset are generally normally distributed without accumulation on either side. The empirical results demonstrate that our data generally satisfy the positivity assumption.

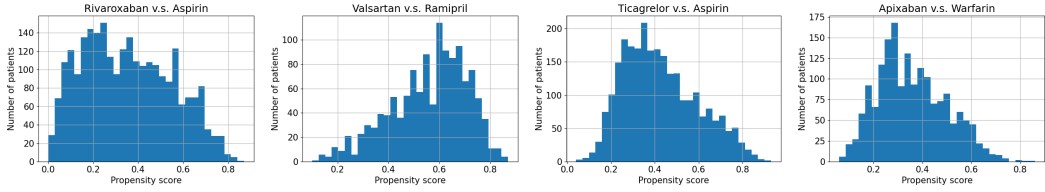

Figure A1: Propensity distribution in each downstream dataset.

**Strong ignorability in our data.**    The strong ignorability assumption should be generally valid in our data. First, the pre-training data have around 3M unlabeled patient sequences with 9,452 unique covariates. With such large patient data and rich covariate space, the model can identify sufficient potential confounders and effectively adjust the confounding bias. Moreover, a recent study Zhang et al. (2022a) of treatment effect estimation further demonstrates that the model trained on tens of thousands of covariates from observational data can even adjust for indirectly measured confounders. Second, according to our study design (as illustrated in Section 3.1 and Figure 2), all the covariates are obtained from the baseline period (time before the target treatment initiation) as potential confounders, and the outcomes are obtained from the follow-up period (time after the target treatment initiation). In this design, all the potential confounders are collected prior to the first target treatment prescription (a.k.a., pre-treatment covariates) thus simple collider bias or mediator bias will be mitigated.

## B    TRANSFORMER ARCHITECTURE

For each single Transformer encoder block, it consists of a multi-head self-attention layer followed by a fully-connected feed-forward layer (Vaswani et al., 2017). The multi-head attention is the most crucial part which can be calculated as,

$$
\begin{aligned}
\text{MultiHead}(\boldsymbol{h}) &= \text{Concat}(\text{head}_1, \ldots, \text{head}_h)W^O; \\
\text{head}_i &= \text{Attention}(\boldsymbol{h}W_i^Q, \boldsymbol{h}W_i^K, \boldsymbol{h}W_i^V) \\
\text{Attention}(Q, K, V) &= \text{Softmax}(\frac{QK^T}{\sqrt{d}})V
\end{aligned}
\tag{1}
$$

where $\boldsymbol{h} \in \mathbb{R}^{d \times d_{\text{model}}}$ denotes the hidden representations and $d$ is the input sequence length. $W_i^Q \in \mathbb{R}^{d_{\text{model}} \times d}$, $W_i^K \in \mathbb{R}^{d_{\text{model}} \times d}$, $W_i^V \in \mathbb{R}^{d_{\text{model}} \times d}$, $W^O \in \mathbb{R}^{nd \times d_{\text{intermediate}}}$ are learnable parameter matrices. $d = d_{\text{model}}/n$ and $n$ is the number of attention heads. We show the detailed model configuration in Fig. A2.

```
"attention_probs_dropout_prob": 0.1,
"classifier_dropout": null,
"hidden_act": "gelu",
"hidden_dropout_prob": 0.1,
"hidden_size": 768,
"initializer_range": 0.02,
"intermediate_size": 3072,
"layer_norm_eps": 1e-12,
"max_physical_time_embeddings": 13,
"max_position_embeddings": 512,
"max_visit_time_embeddings": 361,
"model_type": "bert",
"num_attention_heads": 12,
"num_hidden_layers": 12,
"pad_token_id": 0,
"position_embedding_type": "absolute",
"time_embedding": true,
"torch_dtype": "float32",
"transformers_version": "4.17.0",
"type_vocab_size": 5,
"use_cache": true,
"vocab_size": 9452
```

Figure A2: Model configuration.

# C    ADDITIONAL DETAILS ON EXPERIMENTAL SETUPS

**Pre-training data.**    The pre-training data are obtained from MarketScan Commercial Database [5], which consists of medical and drug data from employers and health plans for over 215 million individuals. In this study, we focus on CAD as the studied disease and stroke as the outcome. The definitions of CAD and stroke are shown in Table A1 and Table A2 respectively.

Table A1: The definition of coronary artery disease (CAD) from observational health data.

| | |
|---|---|
| Reference (PMID) | 16159046, 26524702, 28008010 |
| Criteria | A history of coronary revascularization in the EHR
Or, history of acute coronary syndrome, ischemic heart disease, or exertional angina |
| Diagnostic codes | ICD-9 codes:
410* to 414*
ICD-10 codes:
The best approximation are the following codes:
I20* Angina pectoris
I21* Acute myocardial infarction
I22* Subsequent ST elevation (STEMI) and non-ST elevation (NSTEMI) myocardial infarction
I23* Certain current complications following ST elevation (STEMI) and non-ST elevation (NSTEMI) myocardial infarction (within the 28 day period)
I24* Other acute ischemic heart diseases
I25* Chronic ischemic heart disease |

Table A2: The definition of stroke from observational health data

| | |
|---|---|
| Reference (PMID) | 29202795 |
| Diagnostic codes | ICD-9 codes:
V12.54,
438.0–438.9
ICD 10 codes:
Z86.73
I60-I69
subarachnoid hemorrhage (I60);
intracerebral hemorrhage (I61);
cerebral infarction (I63);
and other transient cerebral ischemic attacks and related syndromes and transient cerebral ischemic attack (unspecified) (G458 and G459) |

---

[5]https://www.ibm.com/products/marketscan-research-databases

**Downstream tasks.**    We demonstrate the flowchart for RCT extraction in Fig. A3. All RCTs are extracted from `https://clinicaltrials.gov/`. We start from 1,593 CAD-related RCTs with stroke as the outcome and end up with 4 RCTs that satisfy all the above criteria. We have included all those 4 RCTs for downstream task construction.

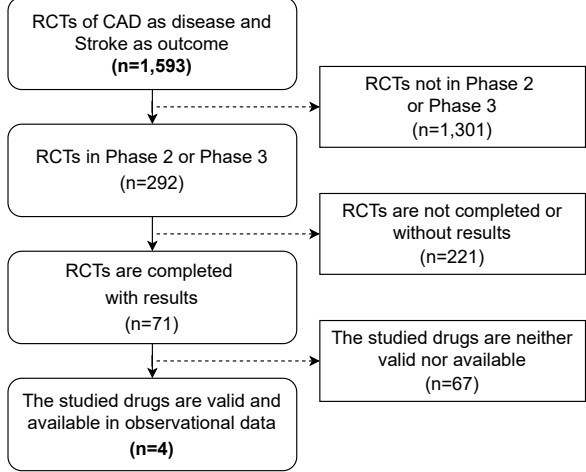

Figure A3: The data flow for RCT extraction. The downstream tasks constructed based on the extracted RCTs.

Table A3: The statistics of the downstream dataset.

| Target v.s. Compared | Rivaroxaban v.s. Aspirin | Valsartan v.s. Ramipril | Ticagrelor v.s. Aspirin | Apixaban v.s. Warfarin |
|---|---|---|---|---|
| # of patients (Target; Compared) | 26340 (9569; 16771) | 12850 (7306; 5544) | 29248 (12477; 16771) | 18187 (6701; 11486) |
| Female (%) | 30.4 | 32.4 | 27.1 | 31.8 |
| Age (group) on index date | 55-64 | 55-64 | 55-64 | 55-64 |
| Patients with stroke (%) | 13.7 | 11.9 | 18.9 | 16.7 |
| Average # of visits per patient | 83.4 | 74.0 | 70.7 | 97.1 |
| Average # of codes per patient | 182.3 | 157.2 | 152.0 | 215.9 |

**Pre-training and fine-tuning data preparation.**    The pre-training is based on large-scale unlabeled patient data, and the fine-tuning is based on small-scale labeled patient data, which are not used for pre-training. We first construct 4 datasets for downstream tasks according to the study design of related randomized clinical trial (RCT). The patients who satisfy the eligibility criteria of the RCT are included in each dataset respectively (see more details of the study design in Fig. 2). Then we construct the unlabeled pre-training data based on all the remaining patients who are not included in any of the 4 downstream datasets. Therefore, the same patients do not appear simultaneously in both the pre-training stage and the fine-tuning stage.

**Evaluation metrics.**    As the true treatment effects are not available in real-world data, we use the influence function-based precision of estimating heterogeneous effects (IF-PEHE) (Alaa & Van Der Schaar, 2019) for model evaluation. Following the same experimental setup, we calculate IF-PEHE as,

- Step 1: Train two XGBoost (Chen & Guestrin, 2016) classifiers for potential outcome prediction denoted by $\mu_0$ and $\mu_1$, where $\mu_a = P(y_a = 1 | X = x)$ using the training set $\mathcal{Z}_{\text{train}}$. Then calculate the plug-in estimation $\widetilde{T} = \mu_1 - \mu_0$ Train a XGBoost (Chen & Guestrin, 2016) classifier propensity score function (i.e., the probability of receiving treatment) $\widetilde{\pi} = P(a = 1 | X = x)$.

- Step 2: Given the estimated treatment effect $\hat{T}(x_i)$ on the test set $\mathcal{Z}_{\text{test}}$, calculate the IF-PEHE with the influence function $\hat{l}$ as,

$$\text{IF-PEHE} = \sum_{x_i \in \mathcal{Z}_{\text{test}}} [(\hat{T}(x_i) - \widetilde{T}(x_i))^2 + \hat{l}(x_i)]$$

$$\hat{l}(x) = (1 - B)\widetilde{T}^2(x) + By(\widetilde{T}(x) - \hat{T}(x)) - W(\widetilde{T}(x) - \hat{T}(x))^2 + \hat{T}^2(x)$$

(2)

where $W = (a - \widetilde{\pi}(x))$, $B = 2a(a - \widetilde{\pi}(x))C^{-1}$, $C = \widetilde{\pi}(x)(1 - \widetilde{\pi}(x))$.

**Implementation details.** The pre-training model architecture follows the BERT$_{\text{base}}$ (Devlin et al., 2018) and most hyperparameters remain the same as default setting. The detailed hyperparameters setup is shown in Table A4 for pre-training , and Table A5 for fine-tuning. With 3 NVIDIA GeForce RTX 2080 Ti 11GB GPUs, the pre-training takes about 20 hours with current setup. We have provided all code in supplemental material.

Table A4: Hyperparameters used in pre-training.

| Parameters | **CURE** |
|---|---|
| Maximum Steps | 100K |
| Initial Learning Rate | 1e-4 |
| Batch Size | 96 |
| Warm-Up Steps | 10K |
| Sequence Length | 256 |
| Dropout | 0.1 |

Table A5: Hyperparameters search space and optimal parameters used for fine-tuning.

| Parameters | Search Space | Optimal Value |
|---|---|---|
| Maximum Epochs | {1,2,3,4,5} | 2 |
| Initial Learning Rate | {1e-5, 3e-5, 5e-5} | 5e-5 |
| Batch Size | {16, 32, 64} | 32 |
| Sequence Length | 256 | 256 |
| Fixed Window Length | 30 | 30 |
| Baseline Window | {90, 180, 360, 720} | 360 |
| Dropout | 0.1 | 0.1 |

Table A6: The parameter size of the proposed method and baselines.

| Method | Model parameters |
|---|---|
| TARNet | 2M |
| DragonNet | 2M |
| DR-CFR | 3M |
| TNet | 4M |
| SNet | 3M |
| FlexTENet | 3M |
| TransTEE | 7M |
| CURE | 93M |

Table A7: The influence of weight ($\alpha$) associated with the discriminator in DragonNet to the model performance on the Valsartan v.s. Ramipril dataset (random seed =42).

| $\alpha$ | AUC | AUPR | IF-PEHE |
|------|-------|-------|-------|
| 0.2 | 0.677 | 0.304 | 0.768 |
| 0.4 | 0.679 | 0.308 | 0.689 |
| 0.6 | 0.680 | 0.310 | 0.660 |
| 0.8 | 0.682 | 0.312 | 0.644 |
| 1.0 | 0.683 | 0.314 | 0.643 |
| 1.2 | 0.679 | 0.315 | 0.593 |
| 1.4 | 0.681 | 0.317 | 0.584 |
| 1.6 | 0.682 | 0.318 | 0.589 |
| 1.8 | 0.683 | 0.319 | 0.586 |
| 2.0 | 0.685 | 0.321 | 0.595 |
| CURE | 0.805 | 0.428 | 0.161 |

# D  ADDITIONAL EXPERIMENTAL RESULTS

**Visualization.**    The self-attention mechanism of the Transformer enables the exploration of inter-action among input covariates. The purpose of visualizing learned attention weights is to better understand our data and interpret what our model learns, how our model makes decisions, and investigate whether the model learns the right associations or not.

As an example, we show the attention weights of a patient from Apixaban treatment group of Apixaban v.s. Warfarin study in Fig. A4. Different colors denote the attention heads and there are 12 heads in total. The medications and diagnosis codes highlighted in the figure are the most related features to the outcome prediction and treatment effect estimation. For example, Amiodarone is an antiarrhythmic medication used to treat and prevent a number of types of cardiac dysrhythmias including atrial fibrillation [6]. A study (Stanifer et al., 2020) shows that apixaban is superior to warfarin in preventing stroke in patients with atrial fibrillation. Those attention weights could be used to analyze the treatment effects in some subgroups that characterized by the attended feature set.

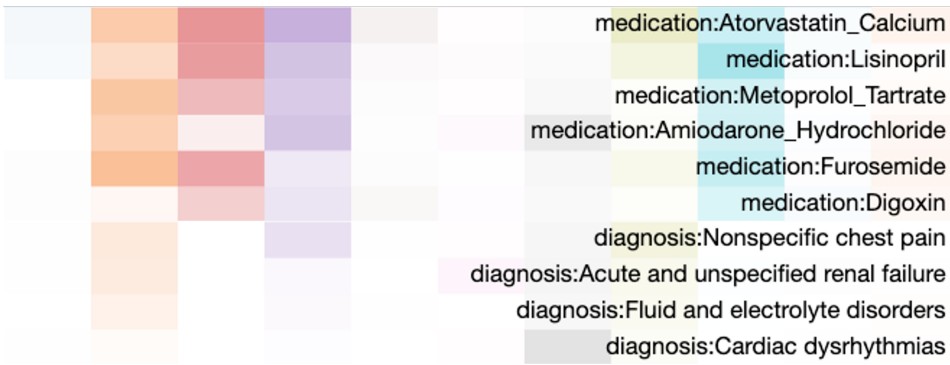

Figure A4: The visualization of Top 10 attention weights associated with the special token `[CLS]` of a patient from Apixaban treatment group.

**Ablation studies in effect of embedding layer.**    We note that in Figure 5, type embedding is helpful to 3 out of 4 datasets as excluding type embedding (w/o. type embedding) yields an increase in IF-PEHE scores in 3 datasets (R v.s. A, V v.s. R and T v.s. A) and a drop in IF-PEHE score in 1 dataset (A v.s. W). According to the basic statistics of 4 datasets (Table A3), we find that the average number of visits per patient and the average number of codes per patient is the largest in A v.s. W dataset among all the 4 datasets. In this case, the type embedding for the data with a larger number of visits/codes may not be as helpful as in the data with a smaller number of visits/codes. And this characteristic is captured by the IF-PEHE metric. Oppositely, excluding type embedding in T v.s. A (which contains the smallest number of visits/codes per patient) yields a significant performance drop (i.e., type embedding is important to T v.s. A dataset). The results demonstrate that each embedding plays a different role in different datasets.

**Ablation studies in effect of downstream data size.**    As shown in Fig. A5, our model can achieve comparable performance as measured by AUPR to the Base Model with only around 5%~10% labeled downstream data.

**Ablation studies in effect of pre-training data size.**    As shown in Fig. A6, the model gradually yields better performance in terms of AUPR scores when the pre-train data size increases.

**Evaluation on non-random assignment.**    We show the t-SNE visualization of learned patient representations of treatment and control group respectively (see Fig. A7). The visualization of our model (CURE) demonstrates that the distribution variance between two groups is marginal and the non-random assignment issue is alleviated.

---

[6] https://www.drugs.com/monograph/amiodarone.html

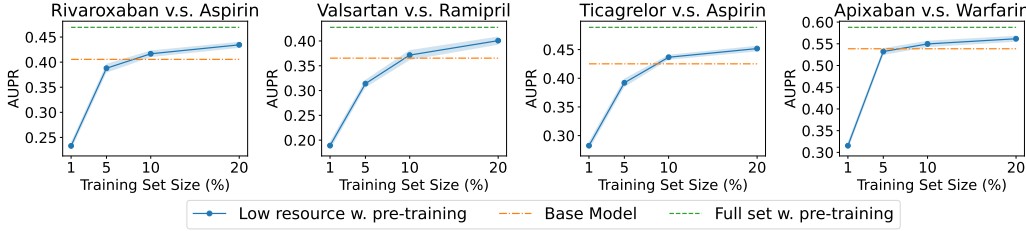

Figure A5: The effect of low resource in fine-tuning datasets on four downstream tasks with different fractions of labeled training set (x-axes). The results are the average of 20 runs.

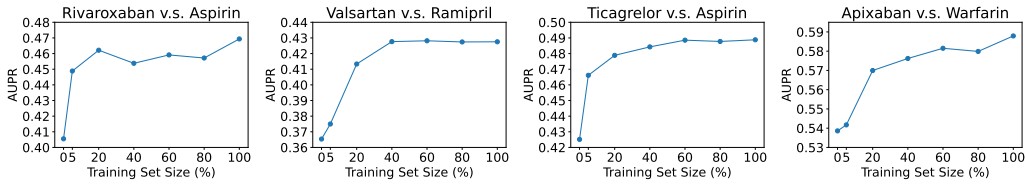

Figure A6: The effect of pre-training data volume on four downstream tasks (average of 20 runs).

We further adapt the design in DragonNet (Shi et al., 2019) in our model's fine-tuning stage. Specifically, we add an additional prediction head for propensity score estimation and modify the loss function to incorporate both outcome prediction and propensity prediction. We compare the new model (CURE+propensity) with the proposed CURE model on 4 downstream tasks respectively. As shown in Table A8, the performance of these two models is comparable in terms of both factual prediction and treatment effect estimation.

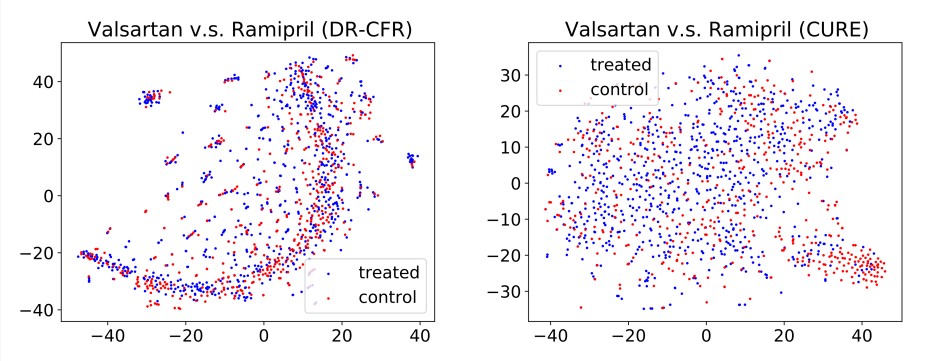

Figure A7: The t-SNE visualization of learned patient representations on Valsartan v.s. Ramipril dataset.

Table A8: Comparison of the proposed model (CURE) and a new model based on CURE but adapting propensity module in DragonNet (Shi et al., 2019) (CURE + propensity) on 4 downstream tasks (random seed=42).

| Method | Rivaroxaban v.s. Aspirin | | | Valsartan v.s. Ramipril | | |
| --- | --- | --- | --- | --- | --- | --- |
| | AUC | AUPR | IF-PEHE | AUC | AUPR | IF-PEHE |
| CURE | 0.786 | 0.419 | 0.186 | 0.805 | 0.428 | 0.161 |
| CURE+propensity | 0.789 | 0.427 | 0.178 | 0.804 | 0.428 | 0.156 |

| Method | Ticagrelor v.s. Aspirin | | | Apixaban v.s. Warfarin | | |
| --- | --- | --- | --- | --- | --- | --- |
| | AUC | AUPR | IF-PEHE | AUC | AUPR | IF-PEHE |
| CURE | 0.807 | 0.502 | 0.211 | 0.838 | 0.597 | 0.182 |
| CURE+propensity | 0.806 | 0.505 | 0.239 | 0.844 | 0.602 | 0.253 |

**Evaluation with RCT conclusion.** We use the bootstrap percentile method which approximates the data sample statistic from the distribution of the bootstrap sample statistic. Specifically, for each downstream data, we first obtain the estimated (individual) treatment effects under 100 random runs with different random seeds. We then calculate the average treatment effects ($c$) of each bootstrap and sort them from smallest to biggest. Finally the (95%) CIs are estimated as $CIs = (c_{0.05/2}, c_{1-0.05/2})$ where $c_\alpha$ denotes the $\alpha \times 100$ percentile of the sorted list.

We evaluate the estimated treatment effects on 100 bootstrap samples. As shown in Table D, the results are consistent with the reported results on 20 bootstrap samples: only a little change in numerical values but no influence on the generated hypothesis.

Table A9: Comparison of the estimated treatment effects with corresponding ground truth RCT. The estimated effects are shown in 95% confidence intervals (CI) under 100 bootstrap runs. The RCT conclusions are obtained from published articles.

| Target v.s. Compared | Estimated Effect (CI) | P value | Generated Hypothesis | RCT Conclusion |
|---|---|---|---|---|
| Rivaroxaban v.s. Aspirin | [-0.009, 0.008] | 0.4140 | No significant difference | No significant difference Anand et al. (2018) |
| Valsartan v.s. Ramipril | [-0.004, 0.011] | 0.0582 | No significant difference | No significant difference Pfeffer et al. (2021) |
| Ticagrelor v.s. Aspirin | [0.019, 0.043] | 5.7814e-25 | T. is less effective than A. | No significant difference Sandner et al. (2020) |
| Apixaban v.s. Warfarin | [-0.037, -0.002] | 1.3257e-15 | A. is more effective than W. | A. is more effective than W. Granger et al. (2011) |

We evaluate the treatment effects estimated by all the baselines and conduct the same hypothesis testing. As shown in Table A10 below, our method correctly generates 3 (out of 4) RCT conclusions that match the ground truth RCT conclusions while the best baselines only identify 2 (out of 4) RCT conclusions.

Table A10: Comparison of the estimated treatment effects with corresponding ground truth RCT of all methods.

| Method | Rivaroxaban v.s. Aspirin Estimated Effect (CI) | P value | Match RCT Conclusion? | Valsartan v.s. Ramipril Estimated Effect (CI) | P value | Match RCT Conclusion? |
|---|---|---|---|---|---|---|
| TARNet | [0.066, 0.095] | 5.678e-10 | No | [-0.037, -0.003] | 0.026 | No |
| DragonNet | [0.18, 0.236] | 5.979e-12 | No | [0.03, 0.07] | 4.681e-05 | No |
| DR-CFR | [0.13, 0.183] | 2.783e-10 | No | [0.002, 0.04] | 0.033 | No |
| TNet | [0.041, 0.07] | 2.509e-07 | No | [-0.038, -0.001] | 0.039 | No |
| SNet | [-0.002, 0.008] | 0.231 | Yes | [-0.051, -0.026] | 3.168e-06 | No |
| FlexTENet | [0.064, 0.108] | 1.529e-07 | No | [-0.079, -0.035] | 3.184e-05 | No |
| TransTEE | [-0.013, -0.002] | 0.018 | No | [-0.019, 0.034] | 0.420 | Yes |
| CURE | [-0.009, 0.006] | 0.452 | Yes | [-0.003, 0.014] | 0.103 | Yes |

| Method | Ticagrelor v.s. Aspirin Estimated Effect (CI) | P value | Match RCT Conclusion? | Apixaban v.s. Warfarin Estimated Effect (CI) | P value | Match RCT Conclusion? |
|---|---|---|---|---|---|---|
| TARNet | [0.064, 0.101] | 2.861e-08 | No | [-0.006, 0.028] | 0.207 | No |
| DragonNet | [-0.013, 0.01] | 0.821 | Yes | [0.018, 0.056] | 6.284e-04 | No |
| DR-CFR | [-0.068, -0.029] | 4.915e-05 | No | [-0.026, -0.002] | 0.047 | Yes |
| TNet | [0.046, 0.069] | 6.474e-09 | No | [0.009, 0.023] | 2.329e-04 | No |
| SNet | [0.005, 0.016] | 4.398e-04 | No | [-0.046, -0.017] | 2.112e-04 | Yes |
| FlexTENet | [0.045, 0.068] | 5.243e-09 | No | [0.012, 0.042] | 0.001 | No |
| TransTEE | [-0.014, -0.009] | 0.0216 | No | [-0.027, -0.002] | 0.027 | Yes |
| CURE | [0.022, 0.040] | 5.982e-14 | No | [-0.039, -0.002] | 4e-04 | Yes |

**Semi-synthetic experiment.** We generate a semi-synthetic dataset based on real patient data obtained from the MarketScan data. Specifically, we simulate treatment assignment $a$ and potential outcome $y$ using pre-treatment covariates $\boldsymbol{x}$ (i.e., historical co-medication, co-morbidities and demographics). The treatment assignment is simulated by $a|\overline{x} \sim \text{Bernoulli}(\text{Sigmoid}(s^T \overline{x} + m))$, where $s \sim \mathcal{N}(0^{|\mathcal{V}|}, 0.1 \cdot I)$, $|\mathcal{V}|$ is the cardinality of medical feature vocabulary, $m \sim \mathcal{N}(0, 0.1)$, $\overline{x}$ denotes the aggregation of all historical covariates. The outcome is simulated by $y|\overline{x}, a \sim \text{Bernoulli}(\text{Sigmoid}(w^T \overline{x} + \beta a + n))$, where $w \sim \mathcal{N}(0^{|\mathcal{V}|}, 0.1 \cdot I)$, $\beta \sim \mathcal{N}(0, 1)$, $n \sim \mathcal{N}(0, 0.1)$.

As we have all potential outcomes under both treatment and control arms available in the semi-synthetic data, the model performance is evaluated with Precision of Estimating Heterogeneous Effects (PEHE), which measures the root mean square error between the true treatment effect and

estimated treatment effect. The comparison results from the semi-synthetic dataset are shown in Table A11. The proposed model CURE yields the best performance among all the baselines. We will add the results for the semi-synthetic dataset in revision.

Table A11: Comparison with state-of-the-art methods on semi-synthetic Valsartan v.s. Ramipril dataset. The results are the average and standard deviation over 20 runs.

| Method | PEHE |
|---|---|
| TARNet | $0.768 \pm 0.012$ |
| DragonNet | $0.759 \pm 0.015$ |
| DR-CFR | $0.714 \pm 0.014$ |
| TNet | $0.784 \pm 0.017$ |
| SNet | $0.776 \pm 0.022$ |
| FlexTENet | $0.791 \pm 0.014$ |
| TransTEE | $0.689 \pm 0.012$ |
| CURE | $\mathbf{0.596 \pm 0.010}$ |

**Additional related work in clinical risk prediction.** Though some existing works Choi et al. (2016b;a); Ma et al. (2017); Che et al. (2018); Luo et al. (2020) in clinical risk prediction using electronic medical records (EHRs) also have patient encoding before sending the input into the model, the patient encoding is unique in our problem scenario and more comprehensive than the existing patient encoding in clinical risk prediction.

Fundamentally, clinical risk prediction is very different from treatment effect estimation in problem formulation, model design, and experiment setup. To the best of our knowledge, we are the first study to incorporate this comprehensive patient encoding with the following pre-training and fine-tuning framework as a whole to representation learning of patient data for treatment effect estimation problems. Even if ignoring the great differences between these two tasks, our patient data encoding is not exactly the same as the encoding methods in existing work in clinical risk prediction (as shown in Table A12). Specifically, besides the token embedding, the proposed patient encoding considers 1) physical time information, which denotes the time gap between the observation and index date, can capture the (irregular) time information; 2) visit time information is crucial to identify the relationship between each medical code and corresponding visit time; 3) type information to help capture the heterogeneity of the medical data (i.e., medications, diagnosis codes and demographics).

Table A12: Patient encoding methods in the clinical risk prediction literature. * Time interval between two observations. ** Position embedding in Transformer Vaswani et al. (2017)

| | Backbone | token (variable) emb. | physical time emb. | visit time emb. | type emb. |
|---|---|---|---|---|---|
| RETAIN Choi et al. (2016b) | RNN | Yes | No | No | No |
| Doctor AI Choi et al. (2016a) | RNN | Yes | Yes* | No | No |
| Dipole Ma et al. (2017) | RNN | Yes | No | No | No |
| GRU-D Che et al. (2018) | RNN | Yes | Yes* | No | No |
| HiTANet Luo et al. (2020) | Transformer | Yes | Yes | Yes** | No |
| Med-BERT Rasmy et al. (2021) | Transformer | Yes | No | Yes | No |
| CURE (Ours) | Transformer | Yes | Yes | Yes | Yes |

To further verify that our proposed embedding gives better performance, We compare our method with GRU-D Che et al. (2018) which is one of the most representative works in clinical risk prediction that explicitly considers the time information. The patient encoding in GRU-D uses the time interval between two observations as the input to the GRU unit in order to capture the time information. We implement the patient encoding method in GRU-D in our model by integrating the token embedding and time interval embedding as the patient embedding and keeping the remaining model architecture the same. We conduct comparison experiments on 4 downstream data respectively. As shown in Table A13, our proposed patient coding method has more than 1.1%, 2.8% and 2.4% respective average AUC, AUPR and IF-PEHE improvement over the patient encoding method in GRU-D.

Compared to the patient encoding method in GRU-D, our proposed patient encoding method is more comprehensive. For time information, we consider both visit time information and physical time information. Here, the physical time information can capture the (irregular) time information and plays a similar role as the time interval embedding in GRU-D. The visit time information, which is not considered in GRU-D, is crucial to identify the relationship between each medical code and the corresponding visit. Besides the time information, we additionally consider the type information to help better capture the heterogeneity of the medical data.

Table A13: Comparison of the patient encoding in CURE with the patient encoding in GRU-D Che et al. (2018) on 4 downstream tasks.

| | Rivaroxaban v.s. Aspirin | | | Valsartan v.s. Ramipril | | |
|---|---|---|---|---|---|---|
| Method | AUC | AUPR | IF-PEHE | AUC | AUPR | IF-PEHE |
| GRU-D | 0.791 | 0.448 | 0.179 | 0.802 | 0.395 | 0.176 |
| CURE | **0.803** | **0.469** | **0.173** | **0.811** | **0.428** | **0.158** |
| | Ticagrelor v.s. Aspirin | | | Apixaban v.s. Warfarin | | |
| Method | AUC | AUPR | IF-PEHE | AUC | AUPR | IF-PEHE |
| GRU-D | 0.784 | 0.458 | 0.244 | 0.813 | 0.561 | 0.248 |
| CURE | **0.793** | **0.489** | **0.198** | **0.826** | **0.588** | **0.224** |

