# OpenReview forum: "CURE: A Pre-training Framework on Large-scale Patient Data for Treatment Effect Estimation"
_ICLR.cc/2023/Conference — Submitted to ICLR 2023_

### Official Review · Reviewer_9BnT · 2022-10-18

**Confidence:** 3
**Correctness:** 3
**Technical Novelty And Significance:** 3
**Empirical Novelty And Significance:** 2
**Recommendation:** 5

**Clarity, Quality, Novelty And Reproducibility:**

I found the paper not to be particularly clear in many aspects, see weakness for the major one.
At the same time the qualiy of the contibution is limited in my humble opinion, a new architecture of the many which could have been designed, proposed and developed, apparently without any explanation or strong motivation for the proposed one.
I found the novelty to be fair and reproducibility to be likely ensured by the availability of code and data, even if the issue of reproducibility is extremely complex to be fairly and consistently assessed.

**Strength And Weaknesses:**

Strength
- the problem tackled is a relevant and urgent one
- results of numerical experiments seems to witness the proposed approach compares favourably to state of the art methods
- the idea of pre-training using unlabled data is useful, even if in a general context it is not new.

Weaknesses
- the issue of missing data is not taken into account, while this is a major one in the case where observational data are taken into account in the health care domain
- some aspects about the assumptions and type of issues that are taken into account are not clear, i.e., I read "The model trained for one particular problem or data may fail to generalize to other scenarios.", what do you precisely mean by generalization? interval validity, external validity ?
- the assumption of positivity is a strong one and I wonder whether it can hold in the practical domain presented by the paper, at least I would like to have seen in the results that the author/s take into account and discuss on this aspect.
- at page 4 when defining x it is not clear which comorbidities are taken into account, those at the beginning of the longitudinal sample or those that have developed through time?
- the assumption that all covariates are confounders or potential confounders is weak and averaging with respect to all covariates, expecially post-treatment covariates is a very bas practice that can greatly affect the performance to compare causal effect of different treatments.
- formula (2) seems to implicitly assume independence between token, type, visit and physical, I think more comments are needed here to justify
- m at the end of page 5 seems to indicate a different quantity tha what presented in previous pages, this make hard to understand the paper.
- I miss the concept of "causal treatment", I know about treatment and causal effect of a treatment but I ignore what a "causal treatment" is, would you be that kind to explain?
- apparently the paper states to be in the conditions to compute counterfactual, this is theoretically impossible unless you provide a theoretical proof of what you state. It is clear I could have missinterpreted what you wrote and if this is the case simply reply by clarifying that you did not intended to say that you can compute counterfactuals with your model. Eventually you can estimate counterfactural, provided a set of assumptions and that you show how this goal is achieved fro a theoretical point of view.
- how to manage the fact that the study population is different from the target population? I think the paper must comment on this fundamental issue
- the results are not discussed in a clear manner and thus not potential justitification or explanation are given to help the interested reader to appreciate the paper
- I read "interaction among input covariates" but I miss what you mean by this, correlation? any other type of association? However, they are not enough to explain anything about the causal story behind the data
- in formula (3) with respect to the fucntion f, you use the argument x while few lines before you used the argument e, thus I'm truly puzzled beacause x and e are defined as different quantities in a previous part of the paper.

**Summary Of The Paper:**

The paper tackles the problem of the estimation of the effect of a treatment with specific reference to the problem of comparing different treatments on given outcomes. The author/s describe/s a framework obtained by combining transformer-based pre-training with fine-tuning by calling it CURE, which is specifically designed for observational data. In particular, the CURE framework is first trained on unlabled data with the aim to obtain a contextual representation of patients, then a fine-tuning step exploits the few available labeled data for comparing the effect of two treatments on a given outcome of interest. The main contributions of the paper, in my understanding, are the sequential encoding devoted to represent longitudinal patient data, an embedding that the paper states to incorporates both structure and time. A second contribution is represened by a set of numerical experiments, togheter with their associated results that, according to what the author/s state/s also share the interesting property of data scalability.

**Summary Of The Review:**

I found the paper to be not easy to read and how it is presented does not make it easy to appreciate the main contributions of the paper itself. The proposed framework, namely CURE, is definitely new, or at least I'm not aware of any other similar framework that has been presented in the specialized literature. However, the paper lacks in clarity and the presented case study has criticalities in terms of the fact that health care data suffer from missingness and several biases, while apparently the paper does not discuss in detail these fundamenatl aspects of the consider domain. Furthermore, also the notation is difficult to follow and in some cases it seem to be a mismatch, e.g. the argument of function f in formula (3). However, results of numerical experiments witness in favour of CURE when compared to state of the art approaches.

---

> ### Author Response · Authors · 2022-11-14
> **Response to Reviewer 9BnT [Q1 - Q3]**
>
> Thank you for your sincere suggestions and insightful feedback! We are glad to receive the encouraging feedback that *“The proposed framework, namely CURE, is definitely new…in the specialized literature”*. We will address the concerns raised in the review by points in the following.
>
> **[Q1 The issue of missing data]**
> First, the observational patient data used in this paper are from a medical claims database. The main purpose of the patient claim records is for future insurance reimbursement so that both insurance companies and physicians are responsible for the claim to avoid any incorrect or missing records otherwise there will be a potential risk for insurance fraud.
>
> Second, even considering the situation of missing data, our proposed pre-train and fine-tune pipeline should be more robust against the missing than the standard machine learning models which are directly trained on specific data/tasks. Pre-training on large-scale patient data can help learn universal and contextualized patient representations. Specifically, the pre-training task (masked language modeling) is to predict the masked positions based on the remaining inputs. Here, intended masked positions can be regarded as missing positions in the data, and thus the pre-training process is to train the model against the missing data.
>
> We understand that missing data is an important topic in the observational study but due to the characteristics of the claims data and proposed pre-train/fine-tune framework, we believe that the missing data issue should be minimal in our scenario. Again, our main focus of this paper is to demonstrate the success of adopting the emerging ML trend of pre-training and fine-tuning to representation learning of patient data for treatment effect estimation
>
> **[Q2 Clarification of “generalization”]**
> The sentence “The model trained for one particular problem or data may fail to generalize to other scenarios.” is to distinguish standard ML practice (i.e., one model for one task) and the emerging trend of pre-training/fine-tuning framework. The generalization of the pre-training/fine-tuning framework here is more related to “external validity” mentioned by the reviewer as 1) there is no overlap between the data used for pre-training and the data used for downstream tasks, and 2) we demonstrate that the model pre-trained on the (unlabeled and large-scale) dataset can be applied to different downstream tasks via fine-tuning the model on (labeled and small-scale) datasets. Since there is a population shift between the pre-training data and downstream data, directly applying the model trained on the pre-training data to the downstream tasks will yield poor performance. Whereas, quick adaption (fine-tuning) can boost the performance on downstream tasks a lot.
>
> For example, to study the comparative treatment effects of Rivaroxaban and Aspirin, the stand ML method is to train a model on the particular dataset of “Rivaroxaban and Aspirin” from scratch and the trained model can be only used for estimating the treatment effects of Rivaroxaban and Aspirin. However, the proposed pre-training and fine-tuning framework will first pre-train the model on unlabeled large-scale patient data (which do not contain any patients in the downstream datasets) and then fine-tune it on different downstream datasets and estimate the treatment effects on each task (e.g., Rivaroxaban v.s. Aspirin, Valsartan v.s. Ramipril, Ticagrelor v.s. Aspirin and Apixaban v.s. Warfarin). **Provided with additional pages, we will make these points clearer in the revised version.**
>
> **[Q3 Positivity assumption”]**
> First, the identification of causal effects from observational data hinges on the imposition of untestable assumptions [Curth et al.,]. The positivity assumptions are not proposed by ourselves but follow several existing works in treatment effect estimation with observational data [Shalit et al., Shi et al., Hassanpour et al., Curth et al.,]. Besides, another type of work on treatment effect estimation with specific application on electronic medical records (EMRs) [Powers et al., Wendling et al.,] also relies on the same assumptions.
>
> Second, we investigate the positivity assumption in each downstream data respectively. Following the positivity evaluation in [Shimoni et al.,], we estimate the propensity score of each individual via a propensity score model. The positivity can be evaluated through the distribution of propensity: whether the distribution is generally smooth with propensities normally distributed instead of much accumulation in either propensity equals 0 or propensity equals 1. As shown in Figure A1 of Appendix D, the estimated propensity scores in each downstream dataset are generally normally distributed without accumulation on either side. **The empirical results demonstrate that our data generally satisfy the positivity assumption.** We have included this discussion in “Positivity in our data” of Appendix A in the revised version.

---

> > ### Author Response · Authors · 2022-11-14
> > **Response to Reviewer 9BnT [Q4 - Q8]**
> >
> > **[Q4 Definition of patient data x”]**
> > According to section 3.1 and study design in Figure 2, the patient data x are demographics, co-medications (prescriptions), and co-morbidities (diagnosis codes) obtained from the baseline period (time before the target/compared treatment initiation) instead of just at the beginning of the observation. **We have clarified the definition in the revision.**
> >
> > **[Q5 Covariates considered as potential confounders”]**
> > According to section 3.1 and study design in Figure 2, we do not consider all covariates along the entire patient observation as potential confounders for adjustment. We only include covariates observed in the baseline period (pre-treatment covariates) as potential confounders and do not include any covariates in the follow-up period (post-treatment covariates). Therefore, our study design alleviates potential collider bias or mediator bias. **We have clarified the definition in the revision.**
> >
> > **[Q6 Encoding in formula (2)”]**
> > The encoding method in formula (2) follows a similar design in the original Transformer model [Vaswani et al.,] and many widely-adopted pre-train models (e.g., BERT model [Devlin et al.,], Roberta [Liu et al.,] and Albert [Lan et al.,]) to realize contextualized embeddings. Adding all the embeddings together does not implicitly assume independence among them. Instead, the adding operation provides a synthesized embedding space that can essentially help the model 1) be more effective in capturing the interaction among different embeddings and automatically attending to important information via self-attention [Wang et al., 2020]; 2) be more efficient in computation as the summed embedding has the same dimension of the original token embedding.
> >
> > In our scenario, the proposed data encoding is to integrate data characteristics from different aspects (e.g., token-level, time-level and type-level). The integrated embeddings are used to learn better patient representations than a single embedding (e.g., only token-level). To verify the effectiveness of the proposed data encoding, we conduct a comprehensive ablation study (Sec. 4.3) to examine the importance of each embedding (e.g., physical time, visit time and type).
> >
> > **[Q7 Notation of masking positions”]**
> > We used another notation ($j$) for masking positions to avoid confusion in the revision.
> >
> > **[Q8 Causal treatment effect]**
> > “Causal treatment effect” refers to the causal effect of a treatment. We also find some papers also use the term “causal treatment effect”: 1) “Does Cox analysis of a randomized survival study yield a causal treatment effect?” [Aalen et al.,] and 2) “Efficient Adjustment Sets for Population Average Causal Treatment Effect Estimation in Graphical Models” [Rotnitzky et al.,]. **We have revised our paper to ensure the definition of this term is clear.**

---

> > > ### Author Response · Authors · 2022-11-14
> > > **Response to Reviewer 9BnT [Q9 - Q13]**
> > >
> > > **[Q9 Estimation of counterfactual]**
> > > Our model is for the estimation of counterfactual and we do not claim in the paper that our model is to “compute counterfactual”. The estimation of counterfactual or treatment effects is based on causal assumptions which are not proposed by ourselves but follow several existing works in treatment effect estimation with observational data [Shalit et al., 2017; Shi et al., 2019; Hassanpour et al., 2019; Curth et al., 2021a; Curth et al., 2021b; Powers et al., 2018; Wendling et al., 2018]. Besides the positivity assumption we illustrate in response to **[Q3 Positivity assumption”]**, we also demonstrate that our model can sufficiently identify and adjust the confounding bias below.
> > >
> > > First, in our paper,  the pre-training data have around 3M unlabeled patient sequences with 9,452 unique covariates. With such large patient data and rich covariate space, the model can identify sufficient potential confounders and effectively adjust the confounding bias. Moreover, a recent study [Zhang et al., 2022] of treatment effect estimation further demonstrates that the model trained on tens of thousands of covariates from observational data can even adjust for **indirectly measured confounders**.
> > >
> > > Second, according to our study design (as illustrated in Section 3.1 and Figure 2),  all the covariates are obtained from the baseline period (time before the target treatment initiation) as potential confounders, and the outcomes are obtained from the follow-up period (time after the target treatment initiation). In this design, all the potential confounders are collected prior to the first target treatment prescription (a.k.a., pre-treatment covariates) thus simple collider bias or mediator bias will be mitigated.
> > >
> > > We recognize that causal assumptions are the fundamentals of treatment effect estimation in observational studies. However, the main focus of this paper is not to thoroughly evaluate and test the validity of each assumption. Instead, we mainly focus on demonstrating the success of adopting the emerging ML trend of pre-training and fine-tuning to representation learning of patient data for treatment effect estimation, together with novel data encoding, necessary transformer architecture modification, and real-world case studies on randomized clinical trials. **We have included this discussion in Appendix A in the revised version.**
> > >
> > >
> > > **[Q10 Study population is different from the target population]**
> > > We are not clear about the “study population” and ”target population“ mentioned in the question. Based on our understanding, the study population refers to the large-scale unlabeled patient data and the target population refers to the small-scale labeled downstream data. There is definitely no overlap between the study population and the target population (i.e., the data used for pre-training do not appear in the downstream data in fine-tuning). The pre-training/fine-tuning framework is proposed to learn universal and contextualized patient representations from the pre-training stage and adapt/fine-tune the pre-trained model to the target population instead of re-training the whole model from scratch every time. According to Figure 6, given only 5% 10% labeled data, the proposed model (CURE) achieves comparable performance to the Base Model which is directly trained on the fully labeled data.
> > >
> > > **[Q11 Results organization]**
> > > According to all the reviews’ comments, we add more results as well as discussions as follows: 1) related literature in clinical risk prediction and empirical comparison of existing patient encoding methods with ours; 2) comparison of the estimated treatment effects with corresponding ground truth RCT under 100 bootstrap samples; 3) discussion and empirical analysis of our data with respect to the causal assumptions; 4) more discussion on the ablation study. We will further improve our result organization according to the reviewer’s comments, if any.
> > >
> > > **[Q12 Interaction among input covariates]**
> > > We mentioned “interaction among input covariates” in the sentence “The self-attention mechanism of the Transformer enables the exploration of interaction among input covariates and provides a potential interpretation of the prediction results.” under the context of data visualization based on self-attention weights. We are not to draw any causal conclusion from the visualization results. Instead, the purpose of visualizing learned attention weights is to better understand our data and interpret what our model learns, how our model makes decisions, and investigate whether the model learns the right associations or not. **We have included this part in “Visualization” of Appendix D in the revision.**
> > >
> > > **[Q13 Notation in formula (3)]**
> > > We revised the typo in the notation that the input of the encoder $f_\theta$ is patient embedding $\boldsymbol{e}$.

---

> > > > ### Author Response · Authors · 2022-11-14
> > > > **Response to Reviewer 9BnT [References]**
> > > >
> > > > References
> > > >
> > > > Shimoni, Yishai, et al. "An evaluation toolkit to guide model selection and cohort definition in causal inference." arXiv preprint arXiv:1906.00442 (2019).
> > > >
> > > > Devlin, Jacob, et al. "Bert: Pre-training of deep bidirectional transformers for language understanding." arXiv preprint arXiv:1810.04805 (2018).
> > > >
> > > > Liu, Yinhan, et al. "Roberta: A robustly optimized bert pretraining approach." arXiv preprint arXiv:1907.11692 (2019).
> > > >
> > > > Lan, Zhenzhong, et al. "Albert: A lite bert for self-supervised learning of language representations." arXiv preprint arXiv:1909.11942 (2019).
> > > >
> > > > Vaswani, Ashish, et al. "Attention is all you need." Advances in neural information processing systems 30 (2017).
> > > >
> > > > Wang, Yu-An, and Yun-Nung Chen. "What do position embeddings learn? an empirical study of pre-trained language model positional encoding." EMNLP (2020).
> > > >
> > > > Shalit, Uri, Fredrik D. Johansson, and David Sontag. "Estimating individual treatment effect: generalization bounds and algorithms." International Conference on Machine Learning. PMLR, 2017.
> > > >
> > > > Shi, Claudia, David Blei, and Victor Veitch. "Adapting neural networks for the estimation of treatment effects." Advances in neural information processing systems 32 (2019).
> > > >
> > > > Hassanpour, Negar, and Russell Greiner. "Learning disentangled representations for counterfactual regression." International Conference on Learning Representations. 2019.
> > > >
> > > > Curth, Alicia, and Mihaela van der Schaar. "Nonparametric estimation of heterogeneous treatment effects: From theory to learning algorithms." International Conference on Artificial Intelligence and Statistics. PMLR, 2021.
> > > >
> > > > Curth, Alicia, and Mihaela van der Schaar. "On inductive biases for heterogeneous treatment effect estimation." Advances in Neural Information Processing Systems 34 (2021): 15883-15894.
> > > >
> > > > Powers, Scott, et al. "Some methods for heterogeneous treatment effect estimation in high dimensions." Statistics in medicine 37.11 (2018): 1767-1787.
> > > >
> > > > Wendling, Thierry, et al. "Comparing methods for estimation of heterogeneous treatment effects using observational data from health care databases." Statistics in medicine 37.23 (2018): 3309-3324.
> > > >
> > > > Zhang, Linying, et al. "Adjusting for indirectly measured confounding using large-scale propensity score." Journal of Biomedical Informatics 134 (2022): 104204.

---

> > ### Comment · Reviewer_9BnT · 2022-11-14
> > **Thx**
> >
> > THank you for replying and addressing the issues I raised, I'll take my time to go through your answers

---

### Official Review · Reviewer_BLi7 · 2022-10-24

**Confidence:** 4
**Correctness:** 3
**Technical Novelty And Significance:** 2
**Empirical Novelty And Significance:** 3
**Recommendation:** 5

**Clarity, Quality, Novelty And Reproducibility:**

See above



**Strength And Weaknesses:**

Strengths:

*The authors provided an abundantly detailed ablation study of the different embeddings used and their effect on the performance of the model.

*The article is well organized. Detailed experiments and analysis.

Weaknesses:

* Given that the benefits of the standard pre-training / fine-tuning approach are well-known, the results in this paper are not particularly novel or surprising.

* Technological innovation is insufficient.

**Summary Of The Paper:**

The authors propose a transformer-based pre-training and fine-tuning framework for TEE from observational data. The proposed method is pre-trained to learn representative contextual patient representations. Then, fine-tuned on labeled patient data for TEE. The proposed method is evaluated on 4 downstream TEE tasks.



**Summary Of The Review:**

The presentation of the paper is clear. It is valuable to study motivation. But technological innovation is limited.

---

> ### Author Response · Authors · 2022-11-14
> **Response to Reviewer BLi7**
>
> Thank you for your sincere appreciation and precious feedback! We are glad to receive positive feedback that *“The article is well organized with detailed experiments and analysis.”* Please see our point-to-point response to your comments below.
>
> **[Q1 The results in the paper are not surprising]**
> In our experiment, we have a baseline model (Base Model) which is directly trained on the downstream datasets using the same architecture as the CURE. As shown in Table 1, the Base Model generally outperforms all other baselines on 4 downstream datasets for both factual prediction and treatment effect estimation. The results demonstrate that the model performance improvement is not only from the advantage of the pre-training/fine-tuning approach itself but also from the proposed patient encoding and transformer model. The patient encoding helps capture the patient information from various aspects (token, time and type) and the transformer model helps model the complex and longitudinal patient sequences. The patient encoding and transformer model as a whole play an important role in learning better patient representations.
>
> **[Q2 The technological innovation]**
> We understand the reviewer’s concern, but we would like to give a different perspective on our methodology. In our study, we purposely aim to make minimal but necessary changes to the transformer architectures from natural language domains to representation learning of patient data. The motivation is inspired by the seminal vision transformer (ViT) [Dosovitskiy et al., 2021] work that demonstrates minimal changes (e.g., image patches and positional encoding) are sufficient to apply existing transformers “as is” to solve computer vision tasks. ViTs are considered as a significant breakthrough in recent years due to the demonstration of a general and unified machine learning framework (i.e., transformers) capable of learning representations for different modalities such as images and texts. Notably, similar to our study, ViT models did not make any “new” methodology nor theoretical guarantees. Similarly, we would (shamelessly) position our proposed model and findings to an identical role of the “ViT model” for the modality of patient data, which is a new territory in AI for Science for machine learning researchers to explore and make contributions. As a promising future direction, our study shows the possibility of developing new multi-modal machine learning models using a unified transformer architecture for processing patient data, images, audios, and texts, which has direct applications to healthcare and beyond.
>
> Moreover, in our study, we have made several novel proposals to adapt the transformers for efficient learning on patient data. These include 1) A new patient data encoding method to encode structured observational patient data. Unlike natural language text, which is inherently encoded as a sequence of words, the patient data need to be preprocessed into a "sequence-like" format before sending to the Transformer encoder. As shown in Fig. 3, we flatten the structured patient data by chronologically going through each medication and diagnosis in each visit and aligning them in one sequence; 2) Incorporating covariate type and time into patient embeddings through a comprehensive embedding layer. Compared to the natural language text, the longitudinal patient data contain a more complex hierarchical structure and are usually irregularly sampled. As shown in Fig. 3, the visit dates are not regularly distributed over time. we propose a more comprehensive embedding layer by including associated code type information and time information (see Fig. 4). Our ablation study in Fig. 5 shows that our improved architecture can significantly improve the performance in terms of all three evaluation metrics (i.e., AUC, AUPR and IF-PEHE) when compared to the standard embedding design adopted in NLP (i.e., token embedding and position embedding), demonstrating that the necessity of proposed changes to transformer architectures.
>
> To summarize, we actually have made several innovations such as the first study of large-scale pretraining in a new domain, novel data encoding, and necessary transformer architecture modification, and our proposed model have demonstrated significant improvements over existing methods. In this regard, we believe novelty should not be a concern.
>
> Reference
>
> Dosovitskiy, Alexey, et al. "An image is worth 16x16 words: Transformers for image recognition at scale." ICLR 2021.

---

### Official Review · Reviewer_wo54 · 2022-10-24

**Confidence:** 4
**Correctness:** 4
**Technical Novelty And Significance:** 3
**Empirical Novelty And Significance:** 3
**Recommendation:** 8

**Clarity, Quality, Novelty And Reproducibility:**

The paper is clearly written and presented. The results should be reproducible given the data.

The novelty rests on the demonstration of training a large transformer-based model as a pre-trained model and fine-tuning it using labeled data for treatment effect estimation.

**Strength And Weaknesses:**

Strengths:
+ Demonstrate the effectiveness of fine-tuning a pre-trained model for patient representation to gain performance improvement
+ Datasets are carefully prepared and the experiments are carefully designed
+ A lot of detailed implementation details are provided in the Appendix
+ BERT is used as the basic architecture, which has been widely adopted and the proposed methodology should be applicable in practice.

Weaknesses:
- Some particular design of time embedding and type embedding is introduced. It is not clear how optimal the design is, and whether there can be alternatives.
- Not much discussion on the issue of selection bias possibly caused by confounder which is important for treatment effect estimation. The use of propensity network is mentioned in the appendix but not in the main text. It seems good to have some related discussion in the main text.

Questions:
- It seems that the results of time embedding only were not reported in Figure 5.
- In term of IF-PEHE, the type embedding seems hurting for three cases and useful for one case (A v.s. W). Any reason?



**Summary Of The Paper:**

This paper proposes to use a large dataset to train a pre-trained BERT-like deep model for patient representation using unsupervised learning approach. It then fine-tine the pre-trained model using data with labels for Treatment Effect Estimation. The motivation is based on the fact that the data with labels is limited. The proposed learning methodology CURE is evaluated based on a carefully prepared claims data and RCT data for treatment effect estimation. Performance improvement is observed based on the proposed CURE, and ablation study is also performed to account for the effectiveness of different tricks used in the proposed CURE framework.

**Summary Of The Review:**

This paper demonstrates the effectiveness of fine-turning a pre-trained model trained using a large unlabelled data for treatment effect estimation. Comprehensive evaluation using a large scale dataset has demonstrated the potential impact of the proposed methodology to be put into practice.

---

> ### Author Response · Authors · 2022-11-14
> **Response to Reviewer wo54 [Q1 - Q2]**
>
> Thank you for your sincere appreciation and valuable feedback! We are delighted to receive the positive feedback that *“the novelty rests on the demonstration of training a large transformer-based model as a pre-trained model and fine-tuning it using labeled data for treatment effect estimation”* and *“the paper is clearly written and presented”*. Please see our point-to-point response to your comments below.
>
> **[Q1 Patient data embedding]**
> The proposed patient encoding method is inspired by the encoding method used in the large-scale pre-train language model BERT [Devlin et al.,]. Instead of directly applying the original embedding design in BERT, we carefully investigate the difference between sequential text data and longitudinal patient data and propose a more comprehensive patient embedding method. Specifically, besides the token embedding, the proposed patient encoding considers 1) physical time information, which denotes the time gap between the observation and index date, can capture the (irregular) time information; 2) visit time information is crucial to identify the relationship between each medical code and corresponding visit time; 3) type information to help capture the heterogeneity of the medical data (i.e., medications, diagnosis codes and demographics).  Empirically, our ablation study shows that the standard embedding design in the language model will be sub-optimal in our scenario (Sec. 4.3 and Figure 5).
>
> Moreover, as suggested by Reviewer w381, we demonstrate the effectiveness of the proposed patient encoding against the existing patient encoding methods in clinical risk prediction problems. We implement the patient encoding method in GRU-D [Che et al., 2018] (which is one of the most representative works in clinical risk prediction that explicitly considers the time information) in our model by integrating the token embedding and time interval embedding as the patient embedding and keeping the remaining model architecture the same. We conduct comparison experiments on 4 downstream data respectively. As shown in Table 1 below, **our proposed patient coding method has more than 1.1%, 2.8% and 2.4% respective average AUC, AUPR and IF-PEHE improvement over the patient encoding method in GRU-D.** The results demonstrate that the proposed patient encoding is optimal compared to the existing patient encoding methods even in other areas instead of treatment effect estimation.
>
> <Table 1>. Comparison of the patient encoding in CURE with the patient encoding in GRU-D [Che et al., 2018] on 4 downstream tasks.
>
> |  | Rivaroxaban v.s. Aspirin |  |  | Valsartan v.s. Ramipril |  |  |
> |---|---|---|---|---|---|---|
> | Method | AUC | AUPR | IF-PEHE | AUC | AUPR | IF-PEHE |
> | GRU-D | 0.791 | 0.448 | 0.179 | 0.802 | 0.395 | 0.176 |
> | CURE | **0.803** | **0.469** | **0.173** | **0.811** | **0.428** | **0.158** |
> |  | Ticagrelor v.s. Aspirin |  |  | Apixaban v.s. Warfarin |  |  |
> | Method | AUC | AUPR | IF-PEHE | AUC | AUPR | IF-PEHE |
> | GRU-D | 0.784 | 0.458 | 0.244 | 0.813 | 0.561 | 0.248 |
> | CURE | **0.793** | **0.489** | **0.198** | **0.826** | **0.588** | **0.224** |
>
> **[Q2 Discussion of selection bias in the main text]**
> Adjusting the selection bias possibly caused by confounder is important for treatment effect estimation from observational data. Following existing work in treatment effect estimation (e.g., S-learner and T-learner [Künzel et al., 2019], and BART [Hill et al., 2011]), our method adjusts the potential selection bias and estimates the potential outcomes via the outcome model given the covariates and treatments. We also implement the selection bias adjustment method in DragonNet [Shi et al., 2019] (one of the most representative works in treatment effect estimation ) in our model.  Specifically, we add an additional prediction head for propensity score estimation and modify the loss function to incorporate both outcome prediction and propensity prediction. The results in Table A8 in Appendix show that the performance of the proposed model (CURE) and the same model but with additional selection bias adjustment (CURE + propensity) is comparable in terms of both factual prediction and treatment effect estimation. **Provided with additional pages, we will move the results and discussion of selection bias in the revised version.**

---

> > ### Author Response · Authors · 2022-11-14
> > **Response to Reviewer wo54 [Q3]**
> >
> > **[Q3 Ablation study in Figure 5]**
> > We note that the legends/notations in Figure 5 may lead to some confusion and misunderstanding. Actually, 1) we include the results of “time embedding only” in Figure 5, which is denoted by “w/o. type embedding” in orange and 2) in terms of IF-PEHE (smaller values denote better performance), type embedding is helpful to 3 out of 4 datasets as excluding type embedding (w/o. type embedding) yields an increase in IF-PEHE scores in 3 datasets (R v.s. A, V v.s. R and T v.s. A) and a drop in IF-PEHE score in 1 dataset (A v.s. W).
> >
> > We further inspect why type embedding is not helpful in A v.s. W. According to the basic statistics of 4 datasets (Table A3 in Appendix), we find that the average number of visits per patient and the average number of codes per patient is the largest in A v.s. W dataset among all the 4 datasets. In this case, the type embedding for the data with a larger number of visits/codes may not be as helpful as in the data with a smaller number of visits/codes. And this characteristic is captured by the IF-PEHE metric. Oppositely, excluding type embedding in T v.s. A (which contains the smallest number of visits/codes per patient) yields a significant performance drop (i.e., type embedding is important to T v.s. A dataset). The results demonstrate that each embedding plays a different role in different datasets. **We have updated these parts in “Ablation studies in effect of embedding layer” of Appendix D in the revised version.**
> >
> > Reference
> >
> > Devlin, Jacob, et al. "Bert: Pre-training of deep bidirectional transformers for language understanding." arXiv preprint arXiv:1810.04805 (2018).
> >
> > Che, Zhengping, et al. "Recurrent neural networks for multivariate time series with missing values." Scientific reports 8.1 (2018): 1-12.
> >
> > Shi, Claudia, David Blei, and Victor Veitch. "Adapting neural networks for the estimation of treatment effects." Advances in neural information processing systems 32 (2019).

---

### Official Review · Reviewer_w381 · 2022-10-26

**Confidence:** 4
**Clarity, Quality, Novelty And Reproducibility:** See below.
**Correctness:** 2
**Technical Novelty And Significance:** 2
**Empirical Novelty And Significance:** 3
**Recommendation:** 5

**Strength And Weaknesses:**

See below.

**Summary Of The Paper:**

This paper proposes a model and pre-training method for estimating conditional expected outcomes as part of causal estimation and evaluates this approach using real and semi-synthetic data.

**Summary Of The Review:**

I am of two minds on this paper. On the one hand, I do not think the proposed method is particularly novel and I think the paper is missing a major chunk of literature on clinical risk prediction. On the other hand, I was very impressed by the experimental setup and comparison to real clinical trials, which I found very thoughtfully done. As this is a machine learning conference and I think the machine learning portion of the paper is lacking, I recommend a marginal rejection.

Major comments:
1. Many, many papers have been written about clinical risk prediction using EHR data, including several that explicitly consider how to encode temporal information (e.g., Che et al. (2018)). I do not think the authors can make the claim that they have developed "a new patient data encoding method to encode structured observational patient data and incorporate covariate type and time into patient embeddings." without reviewing this literature.

2. I do not consider Section 3.1 to be a scientific contribution. These are the preprocessing steps that every EHR-based study must perform and I do not see anything particularly unique about the proposed steps. Please correct me if I have missed something.

3. Notation in Section 2, specifically $y_1(x) - y_0(x)$ is not precisely defined.

4. It is good that the authors mention causal identifiability assumptions in Section 2, but these assumptions cannot be made universally. Rather, for each causal analysis, the authors need to consider the various sources of confounding and censoring and make an argument that there are sufficient observed variables to account for potential biases. No such argument is made for the cases considered.

5. In Section 5.2: 20 bootstrap samples is not enough to reliably estimate 95% confidence intervals. I recommend at least 100, but more would be better. Additionally, there are several ways estimate CIs from bootstrap samples. What method was used here?

---

> ### Author Response · Authors · 2022-11-14
> **Response to Reviewer w381 [Q1]**
>
> Thank you for your sincere suggestions and helpful feedback! We are glad to receive the encouraging feedback that *“experimental setup and comparison to real clinical trials are very impressing”*. We will address the concerns raised in the review by points in the following.
>
> **[Q1 Novelty of proposed patient data encoding]**
> Clinical risk prediction is very different from treatment effect estimation in problem formulation, model design, and experiment setup. To the best of our knowledge, we are the first study to incorporate this comprehensive patient encoding with the following pre-training and fine-tuning framework as a whole to representation learning of patient data for treatment effect estimation problems. Even if ignoring the great differences between these two tasks, our patient data encoding is not exactly the same as the encoding methods in existing work in clinical risk prediction [Choi et al., 2016a; Choi et al., 2016b; Ma et al., 2017; Che et al., 2018; Luo et al., 2020; Rasmy et al., 2021]. We compare the patient encoding methods in the clinical risk prediction with our method in Table 1 below. Specifically, besides the token embedding, the proposed patient encoding considers 1) physical time information, which denotes the time gap between the observation and index date, can capture the (irregular) time information; 2) visit time information is crucial to identify the relationship between each medical code and corresponding visit time; 3) type information to help capture the heterogeneity of the medical data (i.e., medications, diagnosis codes and demographics).
>
> <Table 1>. Patient encoding methods in the clinical risk prediction literature. * Time interval between two observations. ** Position embedding in Transformer [Vaswani et al., 2017]
>
> |  | Backbone | token (variable) emb. | physical time emb. | visit time emb. | type emb. |
> |---|---|---|---|---|---|
> | RETAIN [Choi et al., 2016a] | RNN | Yes | No | No | No |
> | Doctor AI [Choi et al., 2016b] | RNN | Yes | Yes* | No | No |
> | Dipole [Ma et al., 2017] | RNN | Yes | No | No | No |
> | GRU-D [Che et al., 2018] | RNN | Yes | Yes* | No | No |
> | HiTANet [Luo et al., 2020] | Transformer | Yes | Yes | Yes** | No |
> | Med-BERT [Rasmy et al., 2021] | Transformer | Yes | No | Yes | No |
> | CURE (Ours) | Transformer | Yes | Yes | Yes | Yes |
>
> To further verify that our proposed embedding gives better performance,  We compare our method with GRU-D [Che et al., 2018] which is one of the most representative works in clinical risk prediction that explicitly considers the time information. The patient encoding in GRU-D uses the time interval between two observations as the input to the GRU unit in order to capture the time information.  We implement the patient encoding method in GRU-D in our model by integrating the token embedding and time interval embedding as the patient embedding and keeping the remaining model architecture the same. We conduct comparison experiments on 4 downstream data respectively. As shown in Table 2 below, **our proposed patient coding method has more than 1.1%, 2.8% and 2.4% respective average AUC, AUPR and IF-PEHE improvement over the patient encoding method in GRU-D.**  Compared to the patient encoding method in GRU-D, our proposed patient encoding method is more comprehensive. For time information, we consider both visit time information and physical time information. Here, the physical time information can capture the (irregular) time information and plays a similar role as the time interval embedding in GRU-D. The visit time information, which is not considered in GRU-D, is crucial to identify the relationship between each medical code and the corresponding visit. Besides the time information, we additionally consider the type information to help better capture the heterogeneity of the medical data.
>
> <Table 2>. Comparison of the patient encoding in CURE with the patient encoding in GRU-D [Che et al., 2018] on 4 downstream tasks.
>
> |  | Rivaroxaban v.s. Aspirin |  |  | Valsartan v.s. Ramipril |  |  |
> |---|---|---|---|---|---|---|
> | Method | AUC | AUPR | IF-PEHE | AUC | AUPR | IF-PEHE |
> | GRU-D | 0.791 | 0.448 | 0.179 | 0.802 | 0.395 | 0.176 |
> | CURE | **0.803** | **0.469** | **0.173** | **0.811** | **0.428** | **0.158** |
> |  | Ticagrelor v.s. Aspirin |  |  | Apixaban v.s. Warfarin |  |  |
> | Method | AUC | AUPR | IF-PEHE | AUC | AUPR | IF-PEHE |
> | GRU-D | 0.784 | 0.458 | 0.244 | 0.813 | 0.561 | 0.248 |
> | CURE | **0.793** | **0.489** | **0.198** | **0.826** | **0.588** | **0.224** |

---

> > ### Author Response · Authors · 2022-11-14
> > **Response to Reviewer w381 [Q1 (cont'd) - Q4 ]**
> >
> > Moreover, our data encoding is novel with respect to existing widely-adopted pre-trained models (e.g., BERT [Devlin et al., 2018]). Instead of directly applying the encoding method in BERT (though it has achieved great success in many downstream tasks), we thoroughly investigate the difference between sequential text data and longitudinal patient data (i.e., hierarchical structure and irregularity). In order to learn compressive patient embedding, we propose to integrate physical time embedding, visit time embedding as well as type embedding as a whole. To verify the importance of our modified encoding scheme, empirically, our ablation study shows that the standard embedding design in the language model will be sub-optimal in our scenario (Sec. 4.3).  **We have incorporated the literature of clinical risk prediction and empirical analysis in “Additional related work in clinical risk prediction” of Appendix D in the revision.**
> >
> > **[Q2 Notation of potential outcome framework]**
> > Under the causal assumptions (i.e., consistency, positivity and strong ignorability), the potential outcome can be defined as $y_a(\boldsymbol{x})=\mathbb{E}[y|a, \boldsymbol{x}]$, and can be estimated from observational data. **We have added more details in the revision.**
> >
> > **[Q3 Causal assumptions and validation]**
> > First, the identification of causal effects from observational data hinges on the imposition of untestable assumptions [Curth et al., 2021b]. The causal assumptions are not proposed by ourselves but follow several existing works in treatment effect estimation with observational data [Shalit et al., 2017; Shi et al., 2019; Hassanpour et al., 2019; Curth et al., 2021a; Curth et al., 2021b]. Besides, another type of work on treatment effect estimation with applications on electronic medical records (EMRs) [Powers et al., 2018; Wendling et al., 2018] also relies on the same assumptions.
> >
> > Second, in our paper, the pre-training data have around 3M unlabeled patient sequences with 9,452 unique covariates. With such large patient data and rich covariate space, the model can identify sufficient potential confounders and effectively adjust the confounding bias. Moreover, a recent study [Zhang et al., 2022] of treatment effect estimation further demonstrates that the model trained on tens of thousands of covariates from observational data can even adjust for **indirectly measured confounders**.
> >
> > Third, according to our study design (as illustrated in Section 3.1 and Figure 2),  all the covariates are obtained from the baseline period (time before the target treatment initiation) as potential confounders, and the outcomes are obtained from the follow-up period (time after the target treatment initiation). In this design, all the potential confounders are collected prior to the first target treatment prescription (a.k.a., pre-treatment covariates) thus simple collider bias or mediator bias will be mitigated.
> >
> > We recognize that causal assumptions are the fundamentals of treatment effect estimation in observational studies. However, the main focus of this paper is not to thoroughly evaluate and test the validity of each assumption. Instead, we mainly focus on demonstrating the success of adopting the emerging ML trend of pre-training and fine-tuning to representation learning of patient data for treatment effect estimation, together with novel data encoding, necessary transformer architecture modification, and real-world case studies on randomized clinical trials. **We have included this discussion in “Strong ignorability in our data” of Appendix A in the revised version.**
> >
> > **[Q4 Bootstrap samples]**
> > Following the reviewer’s suggestion, we evaluate the estimated treatment effects on 100 bootstrap samples. The results are shown in Table 3 below. The results are consistent with the reported results on 20 bootstrap samples: only a little change in numerical values but no influence on the generated hypothesis.
> >
> > <Table 3>. Comparison of the estimated treatment effects with corresponding ground truth RCT. The estimated effects are shown in 95% confidence intervals (CI) with P value under 100 bootstrap runs.
> >
> > | Target v.s. Compared     | Estimated Effect (CI) | P value    | Generated Hypothesis         | RCT Conclusion                                          |
> > |--------------------------|-----------------------|------------|------------------------------|---------------------------------------------------------|
> > | Rivaroxaban v.s. Aspirin | [-0.009, 0.008]       | 0.4140     | No significant difference    | No significant difference  |
> > | Valsartan v.s. Ramipril  | [-0.004, 0.011]       | 0.0582    | No significant difference    | No significant difference |
> > | Ticagrelor v.s. Aspirin  | [0.019, 0.043]        | 5.7814e-25 | T. is less effective than A. | No significant difference  |
> > | Apixaban v.s. Warfarin   | [-0.037, -0.002]      | 1.3257e-15     | A. is more effective than W. | A. is more effective than W. |

---

> > > ### Author Response · Authors · 2022-11-14
> > > **Response to Reviewer w381 [Q4 (cont'd)]**
> > >
> > > We use the bootstrap percentile method which approximates the data sample statistic from the distribu­tion of the bootstrap sample statistic. Specifically, for each downstream data, we first obtain the estimated (individual) treatment effects under 100 random runs with different random seeds. We then calculate the average treatment effects ($c$) of each bootstrap and sort them from smallest to biggest. Finally the (95%) CIs are estimated as $CIs=(c_{0.05/2}, c_{1-0.05/2})$ where $c_{\alpha}$ denotes the $\alpha\times100$ percentile of the sorted list. **We have included the results with 100 bootstrap samples and details of the bootstrap method in “Evaluation with RCT conclusion” of Appendix D in the revision.**
> > >
> > > Reference
> > >
> > > Devlin, Jacob, et al. "Bert: Pre-training of deep bidirectional transformers for language understanding." arXiv preprint arXiv:1810.04805 (2018).
> > >
> > > Choi, Edward, et al. "Retain: An interpretable predictive model for healthcare using reverse time attention mechanism." Advances in neural information processing systems 29 (2016).
> > >
> > > Choi, Edward, et al. "Doctor ai: Predicting clinical events via recurrent neural networks." Machine learning for healthcare conference. PMLR, 2016.
> > >
> > > Ma, Fenglong, et al. "Dipole: Diagnosis prediction in healthcare via attention-based bidirectional recurrent neural networks." Proceedings of the 23rd ACM SIGKDD international conference on knowledge discovery and data mining. 2017.
> > >
> > > Che, Zhengping, et al. "Recurrent neural networks for multivariate time series with missing values." Scientific reports 8.1 (2018): 1-12.
> > >
> > > Luo, Junyu, et al. "Hitanet: Hierarchical time-aware attention networks for risk prediction on electronic health records." Proceedings of the 26th ACM SIGKDD International Conference on Knowledge Discovery & Data Mining. 2020.
> > >
> > > Rasmy, Laila, et al. "Med-BERT: pretrained contextualized embeddings on large-scale structured electronic health records for disease prediction." NPJ digital medicine 4.1 (2021): 1-13.
> > >
> > > Vaswani, Ashish, et al. "Attention is all you need." Advances in neural information processing systems 30 (2017).
> > >
> > > Shalit, Uri, Fredrik D. Johansson, and David Sontag. "Estimating individual treatment effect: generalization bounds and algorithms." International Conference on Machine Learning. PMLR, 2017.
> > >
> > > Shi, Claudia, David Blei, and Victor Veitch. "Adapting neural networks for the estimation of treatment effects." Advances in neural information processing systems 32 (2019).
> > >
> > > Hassanpour, Negar, and Russell Greiner. "Learning disentangled representations for counterfactual regression." International Conference on Learning Representations. 2019.
> > >
> > > Curth, Alicia, and Mihaela van der Schaar. "Nonparametric estimation of heterogeneous treatment effects: From theory to learning algorithms." International Conference on Artificial Intelligence and Statistics. PMLR, 2021.
> > >
> > > Curth, Alicia, and Mihaela van der Schaar. "On inductive biases for heterogeneous treatment effect estimation." Advances in Neural Information Processing Systems 34 (2021): 15883-15894.
> > >
> > > Powers, Scott, et al. "Some methods for heterogeneous treatment effect estimation in high dimensions." Statistics in medicine 37.11 (2018): 1767-1787.
> > >
> > > Wendling, Thierry, et al. "Comparing methods for estimation of heterogeneous treatment effects using observational data from health care databases." Statistics in medicine 37.23 (2018): 3309-3324.
> > >
> > > Zhang, Linying, et al. "Adjusting for indirectly measured confounding using large-scale propensity score." Journal of Biomedical Informatics 134 (2022): 104204.

---

### Author Response · Authors · 2022-11-18
**Gentle Reminder from Authors**

Dear Reviewers,

Thank you very much for your constructive comments. In the rebuttal phase, we have tried our best to make point-to-point responses based on your comments and improve our submission accordingly. Since the open discussion period will end soon, we would like to send a reminder to take a look at our responses. We are happy to answer any follow-up questions!

Thanks,

Paper 3192 Authors

---

### Decision · Program_Chairs · 2023-01-20

**Decision:**

Reject

**Justification For Why Not Higher Score:**

The paper does not grapple enough with the real-world issues of its application domain, and the ML methods are unsurprising.

**Justification For Why Not Lower Score:**

Thorough work and careful replies to reviewers. One reviewer recommends "accept, good paper."

**Metareview: Summary, Strengths And Weaknesses:**

(a) The paper provides a pretraining method for medical insurance claims data using transformers. The pretraining improves accuracy for downstream causal inference tasks with small subsets of data.

(b) Thorough and well-motivated work; results are believable.

(c) Both the pretraining method and the causal inference do not go beyond putting together standard ML ideas.

This submission is available at https://www.medrxiv.org/content/10.1101/2022.09.09.22279776v1 so it is not anonymous.

Additional comments: This AC has  real-world experience with inference of treatment effects. Despite much discussion, the authors fail to grapple with the difficulties of this. Here are some specific issues:

- Figure A3 shows that out of 71 completed clinical trials, only 4 are eligible for analysis using the authors' method. The real challenge is how to use observational data to replace or confirm the other 67.

- The actual causal inference method used is the most basic possible, a so-called S-learner (Equation 5). This ignores all the reasons to use propensity score weighting and other more sophisticated methods. In the same vein, the authors should discuss "Learning Representations for Counterfactual Inference" from ICML 2016 and follow-on work. That paper shows how to learn representations of medical data that are balanced between the treatment and control group. In contrast, the representation learning here is entirely unsupervised, so the entire burden of adjusting for confounders falls on the S-learner. (Neither the AC nor any reviewer is an author of this 2016 paper.)

- Figure 2 shows how the data subset is selected to mimic an RCT. There is no discussion of issues such as, what if a patient has been prescribed both drugs? In real applications there is always an even more basic question, namely what is the definition of the control group? If the treatment group is patients prescribed aspirin, is the control group all other patients? All others with a certain diagnosis at a certain time?

- The authors are too blithe about the quality of data. They write "both insurance companies and physicians are responsible for the claim to avoid any incorrect or missing records otherwise there will be a potential risk for insurance fraud." But of course there *is* a lot of insurance fraud and even more plain errors in data. What if a patient takes aspirin without a prescription?




**Summary Of Ac-Reviewer Meeting:**

No meeting.

---

> ### Author Response · Authors · 2023-02-17
> **Response to AC's Additional Comments**
>
> We thank the area chair for providing valuable feedback. Based on the metareview, we feel that there are some new questions and comments being raised, and we would like to provide our response for clarification.
>
> **[Q1 Downstream data and related RCTs]** In this paper, we focus on estimating the differences between two treatment options on important outcomes, which is the most widely studied problem in existing work [Shalit et al., 2017; Shi et al., 2019; Hassanpour & Greiner, 2019; Curth & van der Schaar, 2021]. The related clinical trials that study the treatment effect of two drugs are selected for the model evaluation purpose. The remaining unselected 67 clinical trials can be divided into 5 categories according to the type of interventions/treatments: 1) device/procedure (28); 2) biological therapy (5);  3) placebo as compared treatment (23); 4) drug combination (4); 5) drug with specific dosage (7). For 1) and 2), the real-world data (medical claims) does not contain such information. We may verify these trials through other real-world data such as electronic medical records. For 3), we can consult clinicians for helping us define a valid placebo for each target treatment and then estimate the treatment effects of two specific drugs. For 4) and 5), drug combinations/dosages are available and can also be easily incorporated into the modeling process.
>
> **[Q2 Causal inference method]** The choice of simple/sophisticated causal inference methods is orthogonal to our main contribution. In our paper, the method and experiments are designed to investigate and evaluate the proposed novel machine learning pre-training and fine-tuning framework for treatment effect estimation in real-world data. We demonstrate that the proposed minimum but necessary changes to the transformer architecture are effective in the treatment effect estimation. Thus, we adopt a basic method without any complex design for treatment effect estimation during the fine-tuning stage.
>
> **[Q3 Study design for RCT emulation]** We provide more details of the definition of the target treatment group and compared treatment group as follows. Given a target drug (e.g., Rivaroxaban), a patient is assigned to the target treatment group if she/he satisfies 3 criteria: 1) the patient had at least 365 days’ record in the database before/after the first prescription of the drug (i.e., index date); 2) the index date is no prior than the disease initial date; 3) the patient had at least 2 fills of the drug since the index date. The compared treatment group is defined by the same criteria but with respect to the compared drug (e.g., Aspirin). To avoid the overlap between the two groups, we exclude from the compared treatment group any patient prescribed with the target drug.
>
> **[Q4 Missing data ]** Our proposed pre-train and fine-tune pipeline should be more robust against the missing than the standard machine learning models which are directly trained on specific data/tasks. Pre-training on large-scale patient data can help learn universal and contextualized patient representations. Specifically, the pre-training task (masked language modeling) is to predict the masked positions based on the remaining inputs. Here, intended masked positions can be regarded as missing positions in the data, and thus the pre-training process is to train the model against the missing data.